# Anti-Inflammatory and Immunomodulatory Effect of High-Dose Immunoglobulins in Children: From Approved Indications to Off-Label Use

**DOI:** 10.3390/cells12192417

**Published:** 2023-10-07

**Authors:** Francesca Conti, Mattia Moratti, Lucia Leonardi, Arianna Catelli, Elisa Bortolamedi, Emanuele Filice, Anna Fetta, Marianna Fabi, Elena Facchini, Maria Elena Cantarini, Angela Miniaci, Duccio Maria Cordelli, Marcello Lanari, Andrea Pession, Daniele Zama

**Affiliations:** 1Pediatric Unit, IRCCS Azienda Ospedaliero-Universitaria di Bologna, 40138 Bologna, Italy; francesca.conti27@unibo.it (F.C.); angela.miniaci@aosp.bo.it (A.M.); andrea.pession@unibo.it (A.P.); 2Department of Medical and Surgical Sciences, Alma Mater Studiorum, University of Bologna, 40138 Bologna, Italy; anna.fetta2@unibo.it (A.F.); ducciomaria.cordelli@unibo.it (D.M.C.); marcello.lanari@unibo.it (M.L.); daniele.zama2@unibo.it (D.Z.); 3Specialty School of Paediatrics, University of Bologna, 40138 Bologna, Italy; arianna.catelli@studio.unibo.it (A.C.); elisa.bortolamedi@studio.unibo.it (E.B.); 4Department of Maternal Infantile and Urological Sciences, Sapienza University of Rome, 00185 Rome, Italy; lucialeonardi@yahoo.it; 5Department of Pediatrics, Maggiore Hospital, 40133 Bologna, Italy; emanuele.filice@studio.unibo.it; 6IRCCS Istituto delle Scienze Neurologiche di Bologna, UOC Neuropsichiatria dell’Età Pediatrica, 40139 Bologna, Italy; 7Paediatric Emergency Unit, IRCCS Azienda Ospedaliero-Universitaria di Bologna, 40138 Bologna, Italy; marianna.fabi@aosp.bo.it; 8Pediatric Oncology and Hematology Unit “Lalla Seràgnoli”, IRCCS Azienda Ospedaliero-Universitaria di Bologna, 40138 Bologna, Italy; elena.facchini@aosp.bo.it (E.F.); mariaelena.cantarini@aosp.bo.it (M.E.C.)

**Keywords:** anti-inflammatory, children, high dose immunoglobulin, immune dysregulation, immunomodulation, immunomodulatory, inflammation, intravenous immunoglobulin, off-label, pediatric

## Abstract

Background: The large-scale utilization of immunoglobulins in patients with inborn errors of immunity (IEIs) since 1952 prompted the discovery of their key role at high doses as immunomodulatory and anti-inflammatory therapy, in the treatment of IEI-related immune dysregulation disorders, according to labelled and off-label indications. Recent years have been dominated by a progressive imbalance between the gradual but constant increase in the use of immunoglobulins and their availability, exacerbated by the SARS-CoV-2 pandemic. Objectives: To provide pragmatic indications for a need-based application of high-dose immunoglobulins in the pediatric context. Sources: A literature search was performed using PubMed, from inception until 1st August 2023, including the following keywords: anti-inflammatory; children; high dose gammaglobulin; high dose immunoglobulin; immune dysregulation; immunomodulation; immunomodulatory; inflammation; intravenous gammaglobulin; intravenous immunoglobulin; off-label; pediatric; subcutaneous gammaglobulin; subcutaneous immunoglobulin. All article types were considered. Implications: In the light of the current imbalance between gammaglobulins’ demand and availability, this review advocates the urgency of a more conscious utilization of this medical product, giving indications about benefits, risks, cost-effectiveness, and administration routes of high-dose immunoglobulins in children with hematologic, neurologic, and inflammatory immune dysregulation disorders, prompting further research towards a responsible employment of gammaglobulins and improving the therapeutical decisional process.

## 1. Introduction

Immunoglobulin is a plasma-derived medicinal product (PDMPs) sourced from a large pool of healthy blood donors, guaranteeing a broad-spectrum specificity against pathogens, which was employed for the first time in the context of inborn errors of immunity (IEI) as replacement therapy (RT) in 1952 [1,2,3]. The large-scale utilization of immunoglobulins in IEI patients favored the discovery of their key role at high doses (HDs) as immunomodulant and anti-inflammatory therapy in IEI-related immune-dysregulation disorders, such as immune-mediated hematological conditions and rheumatic diseases. This fostered the progressive understanding of the wide-spread biological mechanisms responsible for their multiple therapeutic effects, giving a further boost to HD intravenous immunoglobulin (HD-IVIG) implementation in non-IEI-related immune–mediated neurological and inflammatory disorders. It is true that at the current state of the art, for some of these entities, such as Kawasaki disease (KD), multifocal motor neuropathy (MMN), and Guillain–Barré syndrome (GBS), HD-IVIG represents even the first-choice treatment [3,4].

Immunoglobulins are currently employed according to an approved indication by both the European Medicine Agency (EMA) and the Food and Drug Administration (FDA) [3,4,5,6] in the following pediatric disorders: KD in the acute phase [4,7], primary immune thrombocytopenia (ITP) [3,4,8,9], chronic inflammatory demyelinating polyradiculoneuropathy (CIDP) [4,10] and MMN [4,11]. GBS deserves a separate discussion, as HD-IVIG utilization for its treatment is approved by the EMA but not by the FDA, despite being strongly recommended by evidenced-based medicine in severe forms presenting with an inability to walk independently [12,13,14].

In other pathological conditions, such as rheumatoid arthritis (RA), juvenile idiopathic arthritis (JIA), systemic lupus erythematosus (SLE) and lupus nephritis (LN), catastrophic antiphospholipid syndrome (CAPS), hemophagocytic lymphohistiocytosis (HLH) and multisystem inflammatory syndrome in children (MIS-C), immunoglobulin employment is not sustained by EMA or FDA indications, playing generally a minor but non-negligible role [4].

The multiple mechanisms of action of IVIG rely on its particular two-component structure, in which the Fab variable and the Fcγ constant regions exert distinct functions [15,16] (Figure 1).

In the context of pediatric hematological conditions, in particular ITP, the prominent role probably belongs to immunoglobulin’s Fcγ fragment, which seems to be implied in the positive modulation of the inhibitory Fcγ receptor IIB (FcγRIIB) expression on the phagocytic cells of the reticuloendothelial system. In fact, FcγRIIB binding to IgG autoantibodies prevents them from forming immune complexes with platelets, which would be destined to peripheral destruction by phagocytosis [17,18,19,20].

A further therapeutic immunomodulant action of IVIG is attributable to its saturating action on neonatal Fc receptors (FcRn), involved in the prevention of the clearance of IgG, including pathogenic aAbs, and thus in the prolongation of its half-life, maintaining high levels of disease-causing antibodies, as occurs in ITP [21].

In pediatric rheumatological disorders, such as arthritis, nephritis and myopathy, the mechanism of action is attributable to the Fab region. Specifically, the Fab fragment activates basophils upon interaction with surface-bound IgE, independently from the sialylated Fc/C-type lectin/IL-33 pathway proposed in mice; this signaling results in an enhanced secretion of interleukin (IL-4, IL-6, and IL-8) with a subsequent T helper 2 (Th2)-driven production of IL-4, resulting in the upregulation of the abovementioned FcγRIIB and thus in the rescue of aAbs targeted-cells [22,23,24].

Another Fab region-dependent mechanism in inflammatory myopathies involves IVIG-induced autophagy of hyper-activated reticuloendothelial system cells, upon their IgG endocytosis [25].

Further immunomodulant IVIG functions, in particular in SLE, could involve the dampening of complement hyperactivation by Fcγ fraction opsonization of C3b and Fab region-mediated neutralization of C3a/C5a anaphylatoxins, the accelerated clearance of immune complexes and cellular debris by FcRn saturation, the direct neutralization of pathogenic aAbs and autoreactive B-cells by the IVIG anti-idiotypic Fab fraction ligation with ANCA, dsDNA, BAFF, and APRIL, and the downregulation of the pro-inflammatory interferon cascade through a T helper 1 (Th1)–Th2 imbalance skewed towards a Th2 activation, with a subsequent negative regulation of autoreactive T cells due to decremental IL-2 levels [26,27,28,29,30,31].

Two additional immunomodulatory IVIG functions in rheumatologic diseases, especially in ANCA-associated vasculitis, consists, respectively, of the inhibition of a neutrophil extracellular trap formation in a lactoferrin-dependent manner, upon a FcγR-mediated neutrophil activation [32], and of a Fc segment-independent ROS-mediated cytotoxic effect on eosinophils via a largely caspase-independent pathway involving anti-sialic acid-binding Ig-like lectin-8 (Siglec-8) autoantibodies binding their surface receptor on granulocytes [33].

Eventually, in the setting of hyperinflammatory diseases, such as KD, IVIG impairs NK directly and impair antibody-dependent cell-mediated cytotoxicity activity in a Fc-dependent fashion, enhancing the functions of the NK regulatory cell subset [34]; these effects have been shown also in ITP [35] and CIDP [36,37].

Preliminary studies in the field of neuroinflammation address immunoglobulin’s immunomodulation role in the Fcγ region interaction with neural dendritic cells’ DC-SING, resulting in the modulation of the Th1–Th2–T helper 17 (Th17) differentiation, with an expansion of the regulatory T-cell compartment and strong Th2-driven FcγRIIB expression, at the expense of a Th17 subset’s inhibition.

Other IVIG anti-inflammatory mechanisms in neurological disorders could be due to the impairment of both innate and adaptive immune responses through a reduced Toll-like receptor expression and an ineffective T-cell/B-cell costimulation, respectively [38,39,40,41,42].

In the context of pediatric inflammatory disorders, such as KD and MIS-C, both the Fcγ and Fab fractions exert immunomodulant functions. The Fab region-mediated IVIG binding to self-antigens could dampen the pro-inflammatory vicious circle triggered by the molecular mimicry between viruses and the host tissues responsible for the development of post-viral KD and MIS-C. This prevents cross-linking among self-antigens and pathogenic autoantibodies, as occurs in dermatomyositis with the IVIG Fab fraction, protecting BP180 from degradation fostered by aAbs. Moreover, the FcRn saturation by the Fcγ region could accelerate the clearance of pathogenic aAbs implied in KD and MIS-C [15,43].

IVIG use, mostly off-label, is often limited to multi-refractory conditions as second- or third-line therapy; so, they should be employed as soon as possible in case of multi-refractoriness.

In the off-label context, indications about the optimal dosage, duration, and frequency of immunomodulatory IVIG are usually contingent to expert opinions, varying among different centers; they are also on the basis of the equipe expertise, of the patients’ preferences and comorbidity, and of the environmental affordability. In general, a HD-IVIG protocol ranges from 1000 to 3000 mg/kg of body weight, divided into 400 mg/day for generally 2–5 days [44,45].

In the light of the current imbalance between the gradual but constant increase in the use of immunoglobulins and their availability, we claim the urgency of a thoughtful and responsible utilization of immunoglobulin, according to the latest indications reported by national/international guidelines and reliable reviews.

The largest immunoglobulin consumption is mainly due, in a decrescent order, to acquired (secondary) immunodeficiencies (SID) followed by CIDP and IEI [3,46,47]. Considering the production gap existing between USA and Europe, due in part to different donors’ retention strategies, including non-profit and remunerated donations, the immunoglobulin business, dominated by Big Pharma’s proposal of new therapeutic frontiers and unsupported by high-quality evidence, has become more and more consolidated over the years [4,48,49].

In this background of PDMP’s paucity, our manuscript aims to provide a review of the most recent insights into the major indications of HD-IVIG application in the pediatric context, in order to sensitize healthcare professionals about the responsible employment of this important medical product in a perspective of cost-effectiveness.

## 2. Materials and Methods

### 2.1. Search Strategy

A literature search was performed using PubMed, from inception until 1st August 2023, on HD immunoglobulin use in children, including the following keywords: anti-inflammatory; children; high dose gammaglobulin; high dose immunoglobulin; immune dysregulation; immunomodulation; immunomodulatory; inflammation; intravenous gammaglobulin; intravenous immunoglobulin; off-label; pediatric; subcutaneous gammaglobulin; subcutaneous immunoglobulin. Additional publications were identified and included manually. All types of articles including systematic reviews, case series, and case reports were reviewed.

### 2.2. Eligibility Criteria

Articles were considered for inclusion based on the following criteria:they had to be meta-analysis, systematic reviews, reviews, clinical trials, retrospective and/or prospective observational studies, comparative studies, case series, or reports focusing on the immunomodulatory treatment of hematologic, neurologic, and inflammatory immune dysregulation disorders;the full text had to be available.

On the other hand, articles were excluded if they met any of the following criteria:they were editorials, or conference abstracts;the articles were written in a language other than English;full-text versions of the articles could not be obtained.

### 2.3. Study Selection

The titles and abstracts of the included publications underwent scrutiny by at least two authors. Any conflicts or disagreements that arose during this process were resolved through discussion with at least a third author. For articles deemed likely to meet the eligibility criteria, a comprehensive full-text review was conducted by at least two authors, and in cases of uncertainty, consensus was reached through discussions involving other co-authors, prior to decisions about inclusion or exclusion. In instances where full texts were not readily available through online databases, library requests were made to obtain the necessary documents.

### 2.4. Results

The initial search strategy identified 3279 publications. Following title and abstract review, the full texts of 336 articles were reviewed, resulting in 76 studies fulfilling the inclusion criteria.

Research articles analyzed for the writing of the review are reported in Table 1.

## 3. Results

### 3.1. Immunoglobulins and Immune-Mediated Neurological Disorders

Although HD-IVIG is widely used in pediatric neurological disorders, the literature data on its benefits in these conditions are limited [125]. Available clinical trials are almost exclusively on adults and evidence for children mostly relies on with low sample sizes and restricted end-point measures (Gadian, 2017) [50].

#### 3.1.1. Acute and Chronic Inflammatory Neuropathies

Inflammatory neuropathies are part of a various group of autoimmune diseases affecting the peripheral nervous system in terms of sensory and/or motor complaints, causing significant disability [126,127]. They can have an acute (i.e., GBS in its different variants) or chronic (i.e., CIDP) course, with both a protracted onset and relapsing–remitting or chronic–progressive impairment [128]. MMN is a different entity, which is extremely rare in children, causing non-symmetric arm weakness with no sensorial impairment [128,129]. HD-IVIG use is approved by both FDA and EMA in CIDP (Eftimov, 2013) and MMN (Keddie, 2022) [4,11]. HD-IVIG is the first choice also in GBS, despite only being approved by EMA, based on evidence of faster recovery and reduced morbidity [130,131,132,133,134]. The latest Cochrane meta-analysis about GBS (including both adults and children) found HD-IVIG to be equivalent to plasma exchange (PLEX) in end-point measures, such as time to unassisted walking and ventilation discontinuation (Hughes, 2014) [13]. The common dosage regimen is 2 g/kg dose in one, two or five days. Some studies reported superiority of the 2 day versus 5 day regimen with shorter recovery time [133], others found no difference but a higher relapse rate in the 2 day regimen [132]; a more recent one found the 5 day regimen to lead to faster recovery and a shorter hospitalization time, with no difference in long-term outcome [135].

#### 3.1.2. Myasthenia Gravis (MG)

MG affects the neuromuscular junction, in most cases through specific autoantibodies such those against the acetylcholine (Ach) receptors [54,55]. This results in fatigable ptosis, ophthalmoplegia, dysarthria, dysphagia, and extremity weakness. Ocular symptoms can occur in isolation (ocular MG) or together with systemic symptoms (generalized MG).

Treatment is mainly derived from adult guidelines, given the analogous pathobiological mechanisms and the absence of pediatric clinical trials [136]. Pyridostigmine is usually the first line of therapy, although additive immunosuppressants (corticosteroids, steroid-sparing agents, and anti-B cell agents), and/or thymectomy are often required [136]. HD-IVIG and/or PLEX can induce a prompt short-lasting response in patients with myasthenic crises or severe motor impairment but can also be useful in maintaining remission through monthly cycles, in cases of refractoriness or intolerance to other treatment options [136,137]. In his systematic review, Gadian identified 67 pediatric patients in three studies testing IVIG efficacy in generalized and ocular MG. Efficacy was found in 67% of patients receiving IVIG with or without corticosteroids, 91% with PLEX and IVIG, and 100% with PLEX alone (Gadian, 2017) [50].

#### 3.1.3. Autoimmune Encephalitis (AE)

AE is a heterogeneous group of hyper-inflammatory disorders presenting with an acute or subacute onset of neuropsychiatric symptoms (mental status alteration, seizures, or neurological focal deficits), resulting from an excessive response of the central nervous system (CNS) intrinsic immunity [138,139,140]. Autoantibodies targeting CNS cellular receptors, ion channels, or surface proteins have been increasingly recognized [138]. N-methyl D-aspartate receptor antibody encephalitis (NMDARE) is most frequent in children, although a prevalence of anti-myelin oligodendrocyte glycoprotein (MOG) antibodies emerged in some studies [141,142]. The AE first choice treatment consists of immunomodulant steroids with HD-IVIG or PLEX, followed by second-line immunosuppressants and anti-B lymphocytes agents [143]. Although widely used in clinical practice, all supporting data regarding the safety and efficacy of IVIG in AE come from retrospective case series [130]. The International Consensus Recommendations for the Treatment of Pediatric NMDARE recommend HD-IVIG (or alternatively PLEX) as an additional first line treatment after corticosteroids in severe cases (Nosadini, 2021). First-choice immunotherapy, such as oral corticosteroids or monthly HD-IVIG/intravenous corticosteroids, can be continued for up to 3–12 months, according to severity-based criteria. Refractoriness to first-line therapies is an indication to switch to second-line treatments, to be started 2 weeks after the initiation of first-choice drugs, preferring rituximab and switching to cyclophosphamide or tocilizumab after a further 1–3 months in case of refractoriness. Maintenance therapy beyond 6 months with mycophenolate mofetil or rituximab redosing should be considered in case of severe, relapsing, prolonged and second-line therapy-requiring disease [144].

#### 3.1.4. Autoimmune Demyelinating Disorders of CNS

Acute disseminated encephalomyelitis (ADEM) is a CNS disorder consisting in an immune-mediated (often post-infectious) demyelinization process, resulting in polyfocal neurological deficits associated with encephalopathy and typical reversible white matter lesions at MRI [145]. It mostly occurs in children under 9 years old [146,147]. Anti-MOG antibodies are detected in at least 50% of monophasic and almost all multiphasic forms [145,148]. Observational studies and expert opinions remain to date the only available evidence about the treatment [149,150]. HD intravenous corticosteroids represent the first-choice treatment; HD-IVIG is recommended in the acute phase of monophasic ADEM when corticosteroid therapy fails or is contraindicated (Massa, 2021) [149]. Gadian et al. found HD-IVIG to be associated with complete recovery in 23 out of 26 patients treated (with or without steroids) [50].

The characteristic demyelination pattern of neuromyelitis optica spectrum disorder (NMOSD) includes the II nerve, area postrema, spinal cord, and, rarely, brainstem and diencephalon, and is frequently associated with antibodies against aquaporin-4 (AQP4-NMOSD) [151,152]. A few series (mainly adults’ ones) suggest a potential role of HD-IVIG in AQP4-NMOSD [153,154]. Nevertheless, the increasing and effective use of novel therapies overshadows their use [50].

HD-IVIG is also suggested in multiple sclerosis for acute relapses in the case of incomplete response to HD intravenous corticosteroid, but no systematic study is available (Jancic, 2016) [155,156].

#### 3.1.5. Myelin Oligodendrocyte Glycoprotein (MOG) Associated Disorder (MOGAD)

MOGAD includes different types of demyelinating (such as ADEM, optic neuritis, transverse myelitis, and NMOSD-like phenotype with simultaneous optic neuritis and transverse myelitis [75]), and non-demyelinating (such as meningoencephalitis, encephalitis of the cortex with seizures and of the brainstem, and CNS disorders mimicking vasculitis) manifestations [157,158]. The timely initiation and proper duration of immunotherapy is important, as prompt treatment and corticosteroid therapy longer than weeks have been associated with lower risk of subsequent relapse in a recent study [159]. A pediatric systematic review revealed HD-IVIG is often used after corticosteroids in the acute phase of MOGAD (67%) although there is no randomized controlled trial [55]. Moreover, retrospective data on HD-IVIG use as long-term prophylactic therapy showed that 70% of pediatric patients on monthly HD-IVIG (9–16 months on treatment) were relapse-free [152]. Rituximab, mycophenolate mofetil, or azathioprine can be an alternative as maintenance therapy when used alone, or combined with HD-IVIG (Bruijstens, 2020) [160].

#### 3.1.6. Opsoclonus-Myoclonus-Ataxia Syndrome (OMAS)

OMAS in children usually develops by the first 2 years of life with an acute onset consisting in irritability, ataxia, tremor, myoclonus, drooling, and, later, opsoclonus, defined as chaotic, rapid, multidirectional movements of the eyes in the absence of saccadic intervals. It seems to be mediated by the immune humoral response [141,142], and an underlying tumor (mostly neuroblastoma) is present in about 50% of cases [142]. The prompt Initiation of immunotherapy is indicated because this disorder rarely remits spontaneously, and frequently shows a chronic-relapsing course, with a risk of irreversible neurologic sequelae. A stepwise use of adrenocorticotropic hormone (ACTH) or steroids, PLEX, HD-IVIG, and cyclophosphamide or rituximab is adopted [141]. HD-IVIG, in conjunction with other immunomodulators, is recommended to ameliorate outcomes and reduce further flares in OMAS [50,145]. The adjunction of HD-IVIG in neuroblastoma-associated OMAS treated with cyclophosphamide and corticosteroids showed a better response rate to treatment (81% vs. 41%) in a recent clinical trial [57]. The suggested posology is 2 g/kg distributed across 2–5 days, followed by monthly 1–2 g/kg for up to 12 months (Rossor, 2022) [141].

#### 3.1.7. Other Neurological Disorders with Limited or Inconclusive Evidence

In Rasmussen’s encephalitis [161,162], HD-IVIG use has shown no or time-limited anti-seizure efficacy but seems to have some efficacy in slowing the disease’s progression and reducing the level of disability when used in the early stages of the disease [125,139,163,164].

Immunomodulatory drugs have been increasingly used in febrile infection-related epilepsy syndrome (FIRES), because of proof of cytokine-mediated etiology and minimal response to anti-seizure medications [53,54,165,166,167]. Nevertheless, there is, to date, poor evidence about HD-IVIG effectiveness, limited to a very small percentage of cases [54,168].

In Sydenham’s chorea [92], immunomodulatory therapy did not seem to change the short-term outcome compared with symptomatic therapy alone (i.e., antiepileptics and antipsychotics); nevertheless, individuals without immunomodulatory therapy seemed to have a higher relapse rate [59]. One study demonstrated short-term benefits of HD-IVIG, but data on long-term neurological and psychiatric outcomes were not available [60,61].

In pediatric acute-onset neuropsychiatric syndrome (PANS), HD-IVIG are often considered the preferred treatment [169,170]; nevertheless, a recent meta-analysis revealed a very low grade of evidence of beneficial effects of antibacterial, anti-inflammatory, or immuno-modulating treatments in these patients. Moreover, a moderate grade of evidence of adverse effects was found, particularly related to HD-IVIG (Johnson, 2021) [58].

### 3.2. Immunoglobulins in Hematological Conditions

#### 3.2.1. Immune Thrombocytopenia (ITP)

ITP is a rare immune-mediated acquired bleeding disorder affecting the megakaryocyte lineage of children and adults (the incidence in pediatric population is 3–5 per 100,000/year). ITP is classified as primary when a trigger cannot be identified, or as secondary when it occurs in the context of other pathological conditions (e.g., infections, immunodeficiencies, lymphoproliferative and rheumatological disorders) [171].

In most cases, pediatric patients with newly diagnosed ITP do not have significant bleeding symptoms or other risk factors and may not need any therapy [172].

When a treatment for ITP is required, it is based on the administration of steroids (intravenous HD methylprednisolone or HD dexamethasone), HD-IVIG, and, least commonly, anti-Rh-D immunoglobulin [173]. Second-line treatment options include the monoclonal CD20-antibody rituximab, thrombopoietin-receptor agonists (TPO-Ras) and splenectomy [174].

Numerous studies have shown the effectiveness of HD-IVIG in raising platelet counts in children with ITP, administered alone [69,71,73,74] or in combination with steroids [72,175]. HD-IVIG provides a faster response than steroids, generally increasing the platelet count within 24 h of administration; this is particularly important if bleeding is present in the early stages of the disease. Furthermore, HD-IVIG has been shown to be effective in reducing the development of chronic ITP [69].

In a recent study Mikhail et al. demonstrated that the effectiveness of HD-IVIG in rising platelet count is affected by the initial count, but not by demographic features (i.e., age and sex) [68].

Higashide et al. tried to identify predictive factors for the response to HD-IVIG in newly diagnosed ITP patients, showing that age ≥23 months is the only significant unfavorable factor for long-term response, for which corticosteroid therapy in adjunct with HD-IVIG should be considered as an initial treatment [70].

In the most recent guideline for ITP, the American Society of Hematology recommends administering HD-IVIG in a single dose of 0.8 to 1.0 g/kg, with the possibility of a second dose in unresponsive patients (Provan, 2019) [172].

#### 3.2.2. Autoimmune Hemolytic Anemia (AIHA)

AIHA is a heterogeneous disease characterized by the presence of autoantibodies directed against the patient’s red blood cells. It is very rare in a pediatric setting, with an estimated incidence of 0.8 per 100,000/year [176]. Based on the optimal temperature at which autoantibodies bind erythrocytes and cause hemolysis, AIHA is traditionally classified as warm (IgG-mediated), cold (IgM-mediated) and mixed type.

Warm-AIHA is the most common form in the pediatric population and rarely resolves without treatment [177].

Because of the small number of randomized controlled studies, the treatment of warm-AIHA, especially in children, is still not based on solid evidence [178]. Corticosteroids still represent the first line of treatment, inducing a partial remission in 60–70% of the patients [179]. When steroids alone do not provide sufficient response, second-line treatments, such as splenectomy and immunosuppressive drugs, namely azathioprine, cyclophosphamide, and mycophenolate mofetil, should be considered.

The role of HD-IVIG in the management of AIHA, alone or in combination with prednisone, is controversial and probably linked to its evidence-based effectiveness in ITP and the relatively low-incident adverse effects [65,178,180].

A mixed prospective-retrospective study by Flores et al. demonstrated responsiveness to HD-IVIG in 39% of the 73 patients involved, with an even higher response rate in the pediatric population examined (6 of the 11 children) [64]. Volgaridou et al., in a recent 2021 review, suggest the employment of HD-IVIG (dose: 1 g/kg/day for 2 days) as an additional first-line treatment in children with an inadequate initial response to steroids [177].

#### 3.2.3. Autoimmune Neutropenia (AIN)

AIN is a disease characterized by the presence of circulating autoantibodies targeting neutrophil-specific antigens. Primary autoimmune neutropenia typically affects pediatric patients and generally is self-limited. The neutropenia is classified as a neutrophil count <1.5 × 10^9^/L, but despite the low number of neutrophils, these patients present a mild infectious risk.

In cases where neutropenia is associated with infections, the first line of therapy is represented by the granulocyte colony-stimulating factor (G-CSF) [181]. HD-IVIG is not routinely used in the treatment of AIN in children, but cases of increased neutrophil counts in patients being treated with HD-IVIG are described in the literature [66].

#### 3.2.4. Acquired Hemophilia (AHA)

AHA is a rare coagulopathy caused by the presence of autoantibodies directed against the coagulation factor VIII (FVIII), resulting in an increased risk of bleeding, which can be severe. Generally, AHA occurs in elderly patients with comorbidities, but it has been described in children and postpartum women [182].

Therapeutic options include corticosteroids, cyclophosphamide, cyclosporine, and rituximab.

Despite the presence of several case series describing clinical improvement after immunoglobulin therapy [62,63], HD-IVIG displays a minor role in the management of AHA (Kruse-Jarres, 2017) [183,184].

#### 3.2.5. Neonatal Alloimmune Thrombocytopenia (NAIT)

NAIT is a rare disorder, with an estimated incidence of 1 in 1000 births, caused by the formation of maternal alloantibodies which target fetal platelet antigens inherited from the father. NAIT can result in fetal and neonatal intracranial hemorrhage [185].

Due to the high recurrence risk, in cases of subsequent pregnancies a close surveillance is necessary and, when needed, prophylactic interventions must be taken; the antenatal management of NAIT consists of weekly HD-IVIG infusions during affected pregnancies (Lieberman, 2019) [186].

The first-line treatment of NAIT newborns with <30,000/μL platelets consists of multiple antigen-negative platelet transfusions; in the case of a persistently low platelet count despite transfusions, HD-IVIG can be administered [75].

HD-IVIG is effective in approximately 65% of cases of NAIT [187], although HD-IVIG’s effectiveness seems to be limited when it is applied after antenatal maternal treatment with HD-IVIG [76].

#### 3.2.6. Post-Transfusion Purpura (PTP)

PTP is a severe and rare transfusion reaction, caused by the development of deep thrombocytopenia within 5–10 days of a blood transfusion. In almost all cases, PTP is characterized by the development of antibodies targeting the human platelet antigen 1 on the donor platelets [188].

Possible treatment options include HD-IVIG, systemic corticosteroids, plasmapheresis, and/or rituximab, variously combined. The success of HD-IVIG therapy in slowing platelet destruction in adult patients is stated in many case reports and suggests a possible use in pediatric patients [77,78,79,189].

#### 3.2.7. Thrombotic Thrombocytopenic Purpura (TTP)

TTP is a rare thrombotic microangiopathy, caused by the deficiency of a specific protease targeting the von Willebrand factor, called ADAMTS13. About 10% of all TTP cases occurs in childhood [190].

Microangiopathic hemolytic anemia, consumption thrombocytopenia, and organ injury configure the typical clinical scenario of TTP [191].

The standard treatment of the acute phase of child-onset TTP is based on therapeutical plasma therapy and steroids; in patients unresponsive to first-line therapy, immunomodulatory therapy with rituximab may be considered [190].

The employment of HD-IVIG for the management of TTP, based on few case reports and small-size group series on adult subjects, remains controversial and is currently indicated only in those cases refractory to plasma therapy and in chronic-relapsing TTP (Ding, 2018) [80,81,192].

### 3.3. Immunoglobulins and Inflammatory Diseases

#### 3.3.1. Juvenile Idiopathic Arthritis (JIA)

JIA represents the most frequent rheumatic disease of childhood and gathers heterogeneous conditions. HD-IVIG use for JIA has remained controversial and interest in its efficacy has progressively decreased, largely due to the discovery of new biological treatments in the last 20 years [193]. A trial involving 25 patients suffering from resistant polyarticular JIA demonstrated that the HD-IVIG-treated group experienced a short beneficial effect on clinical symptoms, compared to the placebo [112]. For theoligo- and polyarticular subsets, current guidelines recommend the use of steroidal anti-inflammatory drugs (NSAIDs), corticosteroid intra-articular injection, and disease-modifying anti-rheumatic drugs (DMARDs), especially methotrexate, followed by biological DMARDs in the case of resistant forms (Onel, 2022) [194]. More studies were conducted on systemic JIA (sJIA), a subgroup with visceral manifestations and different treatments [195]. Despite the fact that mild forms could respond to NSAIDs, glucocorticoids and biological therapy with monoclonal antibodies directed against IL-1 and IL-6 are recommended [194]. In the past, off-label HD-IVIG therapy had been attempted. Although in open-label studies an initial improvement in both clinical and laboratory index suggested a potential role in the management of sJIA [113,115,116], Silverman et al. failed to demonstrate statistically significant results between HD-IVIG and the placebo [114]. Therefore, its role in sJIA remained controversial, and the advent of biological therapies limited its application in clinical settings [194,196].

#### 3.3.2. Juvenile Dermatomyositis (JDM)

JDM represents a rare vasculopathy disease, although it is one of the most common inflammatory muscular diseases that affects children. Alongside muscular and cutaneous involvement, clinical features may include systemic symptoms and pulmonary and gastrointestinal manifestations [197]. HD corticosteroids with progressive weaning and the start of immunosuppressive drugs, like methotrexate, represent the mainstay of pharmacologic treatment [198]. In the past, various case series reported that HD-IVIG could potentially be used for JDM, especially in severe, refractory, and steroid-resistant forms [92,94,95,98], as an effective add-on treatment, with improvement in muscle strength, skin manifestations, and steroid-dosage reduction [93,96,97]. In one of the largest investigated cohorts, Lam et al. showed that the HD-IVIG-treated group had comparable or reduced disease than the control patients, and this lasted 4 years after diagnosis, especially for the steroid-resistant patients [91]. More recent clinical trials on adult patients with active dermatomyositis demonstrated better clinical outcomes in those treated with a 16-weeks HD-IVIG course, compared to the placebo group [90].

Current consensus treatment plans include HD-IVIG as a second-line therapy for resistant forms. In particular, current guidelines state that intravenous immunoglobulin may be a useful adjunct for resistant disease, particularly when skin features are prominent (strength of recommendation C) and, in the case of calcinosis cutis, at a monthly dose of 2 g/kg (Enders, 2017) [199,200].

#### 3.3.3. Childhood-Onset Systemic Lupus Erythematosus (cSLE)

cSLE management includes hydroxychloroquine, corticosteroids, and immunosuppressive agents, tailored on disease manifestations and severity. In adult patients, the use of HD-IVIG showed successful results for many complications, like nephritis, hematological and neurological symptoms, vasculitis, and pleuropericarditis [201]. At a pediatric age, HD-IVIG is used as a first-line therapy for hematological abnormalities, especially for SLE-associated thrombocytopenia, although the uncombined use could result in a transient response (Rodriguez, 2017) [202]. A retrospective multicentered study on 215 pediatric patients confirmed a good clinical response to HD-IVIG as a single or combined treatment with other drugs [118]. Some small studies reported the efficacy of prenatal maternal immunoglobulin infusions to prevent cardiac neonatal lupus erythematosus, although the results remained inconclusive [123]. A few case reports on the efficacy of HD-IVIG on neuropsychiatric symptoms [203], like chorea and SLE-associated GBS, can be found in the literature [119,124,204]. A more recent case report showed the successful use of HD-IVIG combined with methylprednisolone and cyclophosphamide for cSLE-associated macrophage activation syndrome (MAS) and neuroimaging alterations [117]. However, given the paucity of well-designed trials, data are not enough to guide strong recommendations (Papachristos, 2020) [205].

#### 3.3.4. Henoch-Schönlein Purpura (HSP)

HSP is a common self-resolving pediatric vasculitis, sometimes associated with gastrointestinal and renal complications for which treatment is debated (Oni, 2019) [206]. Corticosteroids represent the principal therapy in severe forms; HD-IVIG use presented successful results as a second-line option for resistant cases, showing the sustained resolution of bleeding [83,85,89] and of intestinal pneumatosis in one patient [86]. In one case, given the preclusion of steroids because of severe bleeding gastritis, HD-IVIG treatment alone showed a good efficacy on disease flare-up [88]. A recent case–control study, conducted on 64 patients under 18 years of age with gastrointestinal symptoms, refractoriness, or a dependence on glucocorticoids, highlighted a better response to HD-IVIG in younger children [82]. Therefore, pediatric randomized studies are needed to better define its use in this setting. Despite some isolated case reports in adult patients [207], no further trial is available on the role of IVIG in pediatric renal injury prevention and treatment [208]. Regarding purpuric rash, at the present there is no agreement on the treatment of isolated HSP skin symptoms, for which corticosteroids usually represent the first choice, especially in severe hemorrhagic bullous forms [84,209]. Nonetheless, a few cases reporting successful cutaneous healing after HD-IVIG as a second-line drug can be found [84]: in one patient, an improvement in purpuric rash was associated with the resolution of neurological symptoms caused by a brain hemorrhage [87]. However, given the heterogeneity of the published studies on the impact of HD-IVIG on extra-intestinal manifestations, further investigations are needed to evaluate their role.

#### 3.3.5. Kawasaki Disease (KD)

Since the late 1980s, HD-IVIG has been effectively employed as the keystone treatment for KD, significantly modifying the natural course of the disease. Coronary artery aneurysms (CAAs) represent the main KD complication, potentially leading to sudden death, myocardial ischemia, and chronic cardiomyopathy. After the systematic use of HD-IVIG (2 g/kg) during the KD acute phase, in addition to aspirin, the rate of coronary artery aneurysms (CAAs) dropped from 25%, with a mortality rate of 1–2%, to a rate of 3–4% [210] and a mortality rate of 0.1% (McCrindle, 2017) [211]. The precise mechanism of action of HD-IVIG towards KD has not been completely exploited yet [210]. HD-IVIG appears to exert a “generalized” immune-suppressive effect on many levels by (1) neutralizing microbial toxins, (2) suppressing T- and B-lymphocyte activation and favoring their apoptosis in peripheral blood, (3) regulating the Treg/Th17 cell equilibrium, (4) reducing cytokine release, and (5) restoring effector molecules in dendritic cell subsets [212]. However, individual characteristics can vary among KD patients, identifying some children who require supplemental immune-modulatory adjunctive treatments: those patients unresponsive to the first HD-IVIG infusion, and those who are at high risk for developing CAA.

Since HD-IVIG resistance has been reported to correlate with high serum levels of TNF-α, IL-6, and IL-10 and with low levels of IL-5 and eosinophil counts [211], these molecular pathways have been targeted by using corticosteroids, anti-IL-1, anti-TNF-α, or cyclosporin, often with satisfactory outcomes [213].

Treatment with HD-IVIG should be undertaken before day 10 of fever, because it has been shown to decrease the incidence of coronary lesions. However, HD-IVIG can also be infused later, in the case of the persistence of fever and high inflammatory markers. The maximal dose is 2 g/kg, repeatable after 36 h if the fever persists. In the case of HD-IVIG resistance or forms at high risk for coronary involvement, steroids, or biologics, such as anti-IL-1 or anti-TNFα drugs, can be added as second step or directly, as adjunctive first-line therapy (Broderick, 2023) [99].

#### 3.3.6. Multisystem Inflammatory Syndrome in Children (MIS-C)

Since the COVID-19 pandemic, MIS-C has emerged as a severe complication of SARS-CoV-2 infection among children. The clinical and laboratorist analogies between MIS-C and KD led physicians to use HD-IVIG for the treatment of MIS-C since its early appearance, during the acute phase [214,215,216]. HD-IVIG is most frequently associated with glucocorticoids and biologic agents in most severe cases. The association treatment showed satisfactory outcomes in improving cardiac function, CAAs, and the mortality rate [217,218,219,220]. However, no substantial differences were found in terms of inotropic or ventilatory support, or in reduction in disease severity when comparing three different therapeutic regimens: HD-IVIG alone versus steroids alone versus HD-IVIG + steroids [218]. On the other hand, when comparing HD-IVIG + glucosteroid and HD-IVIG, patients’ Intensive Care Unit (ICU) stay was shorter and the need for additional intensification immunomodulatory treatment was lower in the first treatment group [110], as well as the need for inotropic support and the risk for new or persistent cardiovascular dysfunction [109].

Despite there is not an international consensus on MIS-C management, the American College of Rheumatology has confirmed HD-IVIG as its first choice treatment in MIS-C, in association with methylprednisolone (Henderson, 2022) [221].

HD-IVIG acts by multiple mechanisms, one of which seems to induce the death of neutrophils and consequently the clearance of granulocyte-mediated inflammation through PI3K- and NADPH-oxidase-dependent pathways rather than on apoptosis, caspase-1-dependent pyroptosis, necroptosis, or ferroptosis [222,223]. Notably, the reduction in neutrophils counts; IL-1-beta expressing neutrophils were documented in patients receiving HD-IVIG as the unique immunomodulatory treatment. Several markers of inflammation decrease after HD-IVIG [224], which might also block out endothelial and dendritic cell, monocyte and T-lymphocyte functions [225], that are exaggerated in MIS-C.

One complication and continuum of severity of both KD and MIS-C is represented by MAS, a potentially life-threatening variant of hemophagocytic lymphohistiocytosis (HLH) presenting in the context of rheumatological (autoimmune or inflammatory) disorders [226], caused by apoptosis-resistant cytotoxic T-cells and histiocytes, resulting in a massive cytokines release, multi-organ injury, and hemophagocytosis (Ravelli, 2016). Despite the laboratorist features of MIS-C show analogies with MAS, the trend of some markers rise differently [227,228,229], supporting the need for a new definition of MAS in MIS-C. Conflicting data have been reported about HD-IVIG use in MAS, thus its use is still debated: the rationale is supported by its anti-inflammatory action through the inhibition of complement activation, its blocking of antibody Fc fragments and macrophage Fc receptors, and its neutralization of cytokines. Thus, HD-IVIG can be added to steroids or anakinra, but it does not represent the first line of treatment.

HD-IVIG resistance is particularly expressed in clinical patterns complicated by KD Shock Syndrome or MAS, further reflecting a difference in the immunological molecular pathways involved.

In MIS-C myocardial injury can be present in up to 80% of cases [230] and is most likely due to the aberrant systemic inflammation rather than the classic viral myocarditis [231], where the myocyte damage is either directly virus-induced and innate and adaptive immunity-mediated [220]. In addition to these mechanisms, HD-IVIG seems to interfere with the immunological response in myocarditis by disrupting the complement cascade, inhibiting leukocyte adhesion and metalloproteinase release [232] and augmenting anti-inflammatory cytokines with a subsequent reduced production of nitric oxide, which is partly responsible for negative inotropic effects [233]. Despite a Cochrane review which concluded that the evidence of HD-IVIG use in presumed viral myocarditis remains uncertain (Robinson, 2020), if an ongoing infection, or both a post-infectious or non-infectious inflammatory disorder is involved in the pathogenic mechanism, HD-IVIG can be effective, due to its anti-inflammatory, antiviral and immunomodulatory effects [234]. Notably, both the European Society of Cardiology (Caforio, 2013) [235] and the American Heart Association (McCrindle, 2017) [220] encourage HD-IVIG employment due to its immunomodulatory action and the leak of major side-effects, for both viral and autoimmune forms.

### 3.4. Monitoring, Prevention, and Treatment of Adverse Effects

In order to guarantee a better surveillance of major adverse events of IVIG and to prevent minor ones, it is recommended to observe a slow infusion rate, to select the minimum practicable concentration, and to rely on premedication with steroids, antihistamines, and/or NSAIDs, taking into account their potential pro-hemolytic and nephrotoxic effects [236] (Table 2).

B-cell depleting treatments targeting CD19, CD20, B-cell activating factors (BAFFs), and B-cell maturation antigens (BCMAs) play a key role in the context of antibody-mediated autoimmune disorders; in particular, chimeric anti-CD20 rituximab can be employed in rheumatoid arthritis, systemic lupus erythematosus (SLE), and granulomatosis in polyangiitis, humanized anti-CD20 ocrelizumab, and ofatumumab in multiple sclerosis, veltuzumab in immune thrombocytopenia, humanized anti-CD19 inebilizumab in neuromyelitis optica spectrum disorders, and anti-BAFF belimumab in SLE. However, acquired hypogammaglobulinemia is a short- and long-term side effect associated with these monoclonal antibodies [239,248,249,250]. The prevalence of long-lasting hypogammaglobulinemia is conditioned by the type of autoimmune disorder, pre-existent hypogammaglobulinemia, therapy duration, and concomitant adjunctive immunosuppressants [251]. The ever-expanding clinical application of these treatments, along with CD19-targeted chimeric antigen receptor T cell (CAR T) employment in hematological malignancies [252], led to an increase in patients with SID showing susceptibility to encapsulated bacteria and recommended as candidates for prophylactic immunoglobulin replacement therapy (IgRT), at a dose of 400 up to 600 mg/kg every 3–4 weeks, with recommended trough IgG levels ranging from 750 to 850 mg/dL [239,253].

A further relevant issue about gammaglobulins’ employment regards their presence in the commercial preparations of several autoantibodies, whose potential pathogenicity in the recipients, despite being improbable, is currently unknown [254,255,256,257,258].

The recent SARS-CoV-2 pandemic, prompting studies on MIS-C and KD, strengthened the evidence about the fact that temporary anti-Ro52, anti-Ro60, anti-La, and anti-Gastric ATPase IgG antibodies in these patients are attributable to IVIG administration; though, these antibodies show a relative rapid decay under the seronegative cut-off over 2–4 months [259].

A further 2020 study assessed the presence in IVIG preparations of anti-neuronal IgG autoantibodies against glutamic acid decarboxylase (GAD) and aquaporin-4 (AQP4), in titers similar to those of patients with type 1 diabetes and NMOSD, respectively; however, these IVIG-related autoantibodies seem to target only linear epitopes, being part of the natural humoral immune repertoire and, thus, not playing a definite pathogenic role interacting with structural epitopes, as demonstrated by animal passive transfer models [260]. Another detail endorsing the harmlessness of IVIG-related antibodies, in this case of anti-myelin associated glycoprotein ones, is the fact that pathogenic antibodies in demyelinating neuropathies belong prevalently to the IgM isotype [260].

The detection of autoantibodies in patients receiving gammaglobulins can favor, more or less properly, inpatient and outpatient rheumatology and endocrinology consultations. In this context, it is worth considering that low-titer antibodies, such as ANA, anti-Ro, and anti-thyroid antibodies, on the one hand, can also be detected in otherwise healthy people with no clinically manifested disorders, independently from immunoglobulin administration; on the other hand their positivity could be falsified by a high epitope density-driven cross-reactivity at the linear epitope-recognizing enzyme-linked immunosorbent assay dependent on gammaglobulins therapy [261]. So, a standardized approach to attribute pathogenic significance to autoantibodies in immunoglobulin recipients is currently lacking; nonetheless, it is important to consider past laboratory positivity prior to IVIG/SCIG therapy, to employ confirmative structural epitope-recognizing cell-based assays, and to repeat the tests after a time sufficient for the natural decay of these antibodies, being guided by the clinic scenario rather than laboratory features [261].

## 4. Discussion

The review exclusively focused on pediatric indication and use of immunoglobulins to treat inflammatory and autoimmune conditions.

The first reported use of HD-IVIG was described in February 1981 by Imbach et al. on six children, two with idiopathic aplastic anemia and four with refractory ITP, showing ITP resolution within 5–10 days of HD-IVIG therapy, successfully followed with a single infusion every 1–3 weeks [262]. In June 1981, Imbach et al. reported a further seven and six pediatric probands effectively treated with HD-IVIG, without adverse effects, affected by chronic/intermittent and acute ITP, respectively [263]. This prompted the opening of new frontiers in the field of HD-IVIG employment as immunomodulatory and anti-inflammatory therapy, to the extent that Furusho et al. in December 1983 and Fateh-Moghadam et al. in April 1984 published the first case series about the successful use of HD-IVIG in children with KD [107] and young adults with MG [264], respectively.

The very first step towards the discovery of the immunomodulant properties of HD-IVIG was made in 1977 by Péchadre et al., showing a remarkable clinical and electroencephalographic improvement in eight of ten children suffering from severe refractory epilepsy, when treated with large doses of placental gammaglobulins [265].

Since then, several improvements in terms of immunoglobulin production, tolerability, and product availability have led to the large use of HD-IVIG in various pediatric diseases. Although the labelled indications of HD-IVIG are limited, it has been shown to be clinically beneficial in many diseases, displaying a high safety profile which contributed to its large usage, not only in adults but also in pediatric contexts [3,4]. However, especially for HD-IVIG, a deep knowledge of current clinical labelled indications, risks, and benefits should be critically considered.

According to the latest England and Northern Ireland National Immunoglobulin Database Data Report 2020/2021, there has been a general reduction in the yearly number of patients on immunoglobulin therapy for immunologic, neurologic, and infectious conditions (Appendix A) from 2018–2019 to 2020–2021, with a subsequent diminishment in the yearly recorded immunoglobulins volume issued since 2018–2019 [46]. This is in contrast with an ever-expanding demand for immunoglobulins therapy, with a 6–8% annual growth; in particular, SID [3] is currently the leading medical condition for immunoglobulin administration, followed by CIDP and IEIs, according to the latest report of the Australian National Blood Authority, dating back to December 2022 [47].

These considerations fit with the first comprehensive summary of IgG use in Poland over a 5-year period, published in 2023, according to which the neurological and IEI labelled drug programs accounted, respectively, for 34.1% and 21.4% of total IgG consumption, while the remnant 44.5% was employed outside the above-mentioned drug programs, including off-label indications in which HD-IVIG/-SCIG effectiveness is weak, doubtful, or unclear [266].

Similar findings have been reported in the United Kingdom, where SID appears to be driving an increased demand for IgRT [3]. It is worth noting that recommendations for the treatment of SID are not comprehensively defined or strictly specified and vary globally [267,268,269].

Recent updates from the EMA have revised the indications for IgRT in SID. According to the updated guidance, IgRT is recommended for all patients experiencing severe or recurrent infections, those with ineffective responses to antimicrobial treatments, and individuals presenting with either proven specific antibody failure or serum IgG levels below 4 g/L. Proven specific antibody failure is specifically defined as the inability to achieve at least a twofold increase in IgG antibody titer, in response to pneumococcal polysaccharide and polypeptide antigen vaccines [267].

It is worth noting that the supporting evidence from large controlled clinical trials to substantiate these EMA indications for SID treatment is currently lacking. So, there is an urgent need for further research, aligning with the call for the harmonization of existing clinical practices, as there are discrepancies in guidelines across European countries with respect to the initiation, dosing, and discontinuation of IgRT in SID [270]. To address these disparities and provide guidance in areas where there is uncertainty or variation, expert opinions have been offered and referenced in a 2023 European expert question–answer-based review on IgRT employment in SID (Cinetto, 2023) [271].

Despite the intravenous route being, to date, the only administration process with FDA approval for the use of HD in inflammatory and autoimmune conditions, there is an ever- and ever-increasing SCIG utilization in clinical practice, especially for immunoglobulins enriched with human recombinant hyaluronidase, allowing higher infusion volumes and a quicker administration compared to the other products [272].

It is a worthy reminder that, except for KD [4,7], ITP [3,4,8,9], CIDP [4,10], MMN [4,11] and GBS [12,13,14], HD-IVIG is currently off-label for all the other immune-dysregulation disorders, such as RA, JIA, SLE, LN, CAPS, HLH, and MIS-C, playing generally a minor but non-negligible role [4].

Furthermore, in the above-mentioned pediatric disorders, HD-IVIG utilization is preferred over corticosteroids and monoclonal antibodies, not only for its evidence-based medicine effectiveness but also for its minor systemic adverse effects and greater availability and feasibility [4].

The COVID-19 era accelerated the process that had already begun in the 1990s in the United Kingdom and in the 2000s in the United States, when rapid infusion of SCIG was first introduced, gradually replacing the intravenous route by reason of its clinical advantages [273,274,275,276,277,278]. Indeed, SCIG guarantees steady plasmatic immunoglobulin G levels, elevated tolerability, and less systemic adverse effects than IVIG, a more flexible self-made administration with a subsequent better adherence to therapy, a higher quality of life, and a reduction in hospital access, which in turn leads to a decrease in infectious exposures, which is potentially dangerous, especially for patients with IEI and SID on IgRT [272]. Furthermore, the shift from IVIG to SCIG is boosted by the subcutaneous route-related economic advantages and cost reductions demonstrated by several research studies, in the face of major and minor studies assessing the non-inferiority and even the superiority of SCIG in respect to IVIG [272,279,280,281,282].

The ongoing shift from IVIG to SCIG is witnessed also by the aforementioned comprehensive summary, published in 2023, of IgG use in Poland over a 5-year period, according to which, in 2020, 78% of Polish children with IEI received SCIG [266].

From an economic standpoint, there is growing evidence to support the potential benefits of lifelong SCIG treatment. For instance, a cost minimization model, assessing the expenses of IVIG versus SCIG from the perspective of the Spanish National Healthcare System, assumed that all IVIG infusions were hospital-administered, while 95% of SCIG infusions were carried out at home [283]. These economic models, applied also in other European and American healthcare systems, suggest that SCIG could be a cost-effective alternative to IVIG, not only in Spain but also in Switzerland [284,285], Canada [286,287] and Pennsylvania [288], both for neurological and IEI labelled drug programs.

These considerations endorse the fact that, during an acute event, although IVIG availability could be more immediate, it is important to consider the use of SCIG, especially in view of the relatively low immunoglobulin volumes needed. This is, first of all, because the subcutaneous route is technically more feasible, particularly in long-term treatment (e.g., myasthenia gravis), which is frequently complicated by a depletion of the venous heritage; secondly, this is due to the fact that the amount of immunoglobulins needed in children, being weight dependent, is lower, resulting in cost and product savings.

On the other hand, the ever-increasing strictness of criteria to be considered a suitable donor [4,289], along with the SARS-CoV-2 pandemic, has led to a decrease in blood and plasma donation [278], resulting in difficulty finding the immunoglobulins in general, with a poor availability of the subcutaneous product in particular. Indeed, in the pandemic era, immunoglobulin use was reserved mainly for patients affected by predominantly antibody deficiencies, undergoing IgRT as a first priority.

Autoimmune and inflammatory conditions in the pediatric population may underlie a predisposing factor related to an unbalanced immune system; the severe and acute onset of symptoms often justify an off-label employment of HD-IVIG. Indeed, in selected cases, it should be important to check the immunological status before immunoglobulin is used, considering inflammatory/autoimmune disorders as a potential red flag for an IEI [290,291].

Considering that HD-IVIG requires four to five times the dose employed in patients on IgRT, further studies to validate the standardized protocols, while reconsidering the benefits, risks, cost-effectiveness, and different administration routes of immunoglobulins, for children with immune-dysregulation disorders, are needed to guide clinicians towards a thoughtful and responsible employment of this important medical product and to improve the therapeutical decisional process. The field is waiting for the development and validation of alternative and/or complementary immunomodulatory therapies reducing the demand for HD immunoglobulins [3], such as FcRn inhibitors (nipocalimab, rozanolixizumab, batoclimab, and efgartigimod), which are presently in clinical trials for CIDP and MG treatment [292,293]; an anti-CD19 antibody provoking B-cell exhaustion (inebilizumab) and an anti-IL-6 receptor antibody affecting the B- and T-cell maturing process (satralizumab), which are, to date, under investigation for use in NMOSD [294,295]; the proteasome inhibitor bortezomib, anecdotally used for pediatric refractory ITP [296]; and novel complement inhibitors (zilucoplan and ravulizumab), at present in clinical trials for subjects with MG [297].

In response to a rising IVIG/SCIG demand, the strategies to apply in order to ensure consistency and coherence in IVIG/SCIG utilization practices could consist of:Formulating international guidelines on which a spectrum of other global, national, regional, and local initiatives should be aligned, encompassing consensus documents, audit programs, and monitoring frameworks.Implementing robust data collection mechanisms for IVIG/SCIG usage, IVIG/SCIG-requiring disease prevalence and diagnoses (e.g., IEI registries), as well as dosage and clinical outcome data, thereby facilitating an evidence-based evaluation of the efficacy of IVIG-SCIG prescriptions.Exploring viable alternatives to IVIG/SCIG therapy to mitigate the growing demand and address supply constraints.Ensuring access to proficient immunologists who possess expertise in the judicious prescription of IVIG/SCIG, while also fostering collaboration among less experienced clinicians and their more seasoned counterparts to facilitate knowledge sharing and skill development.Refraining from utilizing IVIG/SCIG in cases where its efficacy is tenuous, uncertain, or not well-established, unless the situation involves a life-threatening condition that may potentially benefit from rescue HD-IVIG. Whenever possible and if suitable, it is preferable to explore alternative, safe, and cost-effective therapeutic options. This approach is aimed at preserving the availability of IVIG/SCIG for patients with IEI primarily suffering from antibody deficiencies, for whom IgRT serves as a crucial and life-saving treatment.

## Figures and Tables

**Figure 1 cells-12-02417-f001:**
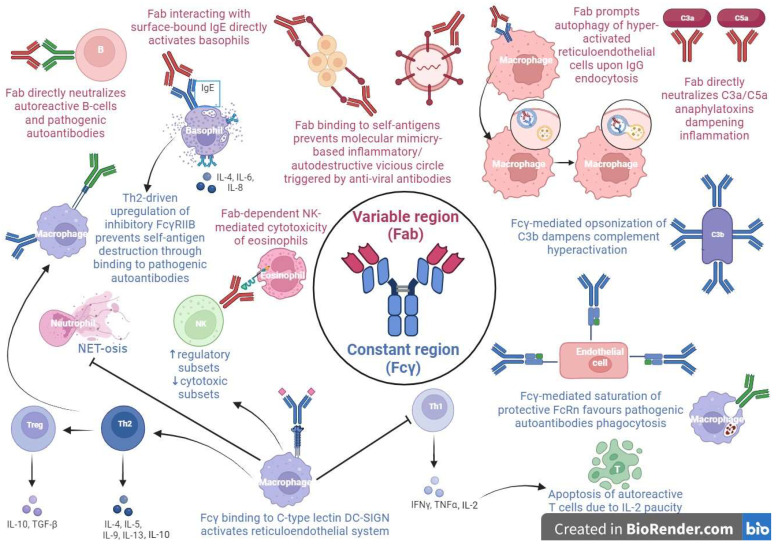
The different mechanisms of action of high dose intravenous immunoglobulins (HD-IVIG) as anti-inflammatory and immunomodulatory agents.

**Table 1 cells-12-02417-t001:** Research articles involving children assessing the anti-inflammatory and immunomodulatory effect of high-dose immunoglobulins (HD-IVIG).

Author	Year	Type of Study	Objective	Outcome
Neurologic Conditions				
Acute disseminated encephalomyelitis (ADEM)				
Gadian J et al. [50]	2017	Systematic review without meta-analysis	Comparison between IVIG and corticosteroids ± plasma exchange (PLEX)/ supportive care no treatment in terms of complete recovery	IVIG and steroids can improve recovery in initial treatment of ADEM
Vitaliti G et al. [51]	2015	Systematic review without meta-analysis	Effects of corticosteroids in terms of complete recovery	IVIG is a possible treatment for ADEM, especially for the steroid-resistant/dependent form, although there is insufficient evidence about its effectiveness
Chronic inflammatory demyelinating polyneuropathy (CIDP)				
Racosta JM et al. [52]	2017	Systematic review with meta-analysis	Comparison between IVIG and subcutaneous (SCIG) immunoglobulin in terms of symptoms [baseline change in the Medical Research Council Sum Score (MRC-SS)], efficacy, safety, adverse effects, and conditions of use for the different treatments	No statistically significant difference between IVIG and SCIG on the MRC-SS for muscle strength
Gadian J et al. [50]	2017	Systematic review without meta-analysis	Comparison between IVIG and PLEX/corticosteroids in terms of sensory and motor function, assessed by the Modified Rankin Scale, neurological and/or electrophysiological tests	IVIG is a potential efficacious first-choice therapy for chronic inflammatory demyelinating polyneuropathy (78% rate of response), when compared with corticosteroids (70% rate of response) and PLEX (14% rate of response) in terms of sensory and motor function
Febrile infection-related epilepsy syndrome (FIRES)				
Gaspard N et al. [53]	2018	Critical review and invited commentary	The First International Symposium regarding new-onset refractory status epilepticus (NORSE) and FIRES to improve patient management	Among 225 patients with FIRES, positive effects were reported in 11/63 (17%) treated with steroids, 5/94 (5%) with IVIG, 19/35 (54%) with ketogenic diet, 2/18 (11%) with plasmapheresis, 3/5 (60%) with hypothermia, and 1/3 (33%) with rituximab
Kramer U et al. [54]	2011	Retrospective multicenter study	To search for the correlations between different therapies (IVIG, steroids, barbiturate, ketogenic diet) and outcome in 77 children with FIRES	IVIG, ketogenic diet, and a barbiturate-induced coma were the only efficacious treatment modalities for FIRES in two, one, and one patients, respectively
Guillain–Barré syndrome (GBS)				
Gadian J et al. [50]	2017	Systematic review without meta-analysis	Comparison between IVIG and PLEX/supportive care/no treatment in terms of time to improvement/recovery	IVIG speeds up recovery, reducing time until function improvement, but does not affect maximal disability score
Vitaliti G et al. [51]	2015	Systematic review without meta-analysis	Comparison between IVIG and PLEX/supportive care/no treatment in terms of time to improvement/recovery	IVIGs are a possible therapy for GBS in comparison with PLEX or supportive care
Hughes RA et al. [13]	2014	Systematic review with meta-analysis	Explore the differences between IVIG and PLEX/placebo/no treatment in terms of time to recovery and improvement in disability grade 4 weeks after the randomization	Addition of IVIG hastens recovery, significantly improving disability grade 4 weeks after randomization compared with supportive care alone
Myasthenia gravis (MG)				
Gadian J et al. [50]	2017	Systematic review without meta-analysis	Comparison between IVIG and corticosteroids in terms of clinical response to treatment	IVIG may improve responses in patients with MG, although the evidence about itstheir effectiveness in the disease’s exacerbation is low
Myelin oligodendrocyte glycoprotein (MOG) antibody-associated disease (MOGAD)				
Klein da Costa et al. [55]	2021	Systematic review	To perform a review on treatment of children with MOGAD	The most common acute-phase treatments for MOGAD are intravenous corticosteroids and IVIG Relapsing disease course is the main indication for long-term treatment Corticosteroids, rituximab azathioprine, and mycophenolate mofetil are the most common maintenance treatments Observational studies show less relapses with periodic infusion of IVIG and rituximab
N-methyl D-aspartate receptor antibody encephalitis (NMDARE)				
Nosadini et al. [56]	2021	Meta-analysis	To assess the safety of immunotherapies in individuals with NMDARE; to identify early predictors of relapse and poor functional outcome; to assess variations in immunotherapy employment and disease course since NMDARE was first reported 14 years before	Treatment factors at first event significantly associated with a good functional outcome 12 months from disease onset included: - first-line treatment with therapeutic apheresis alone; - corticosteroids in combination with IVIG; - corticosteroids in addition to IVIG and apheresis Rituximab and long-course IVIG significantly correlated with non-relapsing disease course
Gadian J et al. [50]	2017	Systematic review without meta-analysis	To evaluate effectiveness of IVIG, with or without adjunctive immunomodulatory therapies, in terms of recovery in patients with NMDARE	IVIG may improve recovery in NMDARE IVIG or other immunomodulators may determine a better outcome in NMDARE (grade C)
Opsoclonus-myoclonus-ataxia syndrome (OMAS)				
de Alarcon PA et al. [57]	2018	A randomised, prospective, open-label, phase III therapeutic trial	To evaluate the response to prednisone and check if IVIG adjunction further ameliorates response in 53 pediatric patients with neuroblastoma-related OMAS	IVIG in adjunction to prednisone significantly ameliorate OMAS response rate IVIG is the basis of additional therapy
Gadian J et al. [50]	2017	Systematic Review without meta-analysis	To evaluate the effectiveness of IVIG, steroids, and rituximab/cyclophosphamide in comparison with corticosteroids/ACTH alone in terms of recovery in patients with opsoclonus myoclonus syndrome	IVIG may improve recovery in opsoclonus myoclonus syndrome in combination with corticosteroids/ACTH; early therapy intensification with rituximab/cyclophosphamide adjunction may improve recovery It is possible that IVIG in combination with other immunomodulators improve outcome in opsoclonus myoclonus syndrome (grade C)
Pediatric acute-onset neuropsychiatric syndrome (PANS)				
Johnson M et al. [58]	2021	Systematic review without meta-analysis	To assess effects of treatment in children with PANS	The certainty of evidence for beneficial effects and for adverse effects of immunomodulating IVIG and PLEX is low and moderate, respectively, in patients with PANS Potential beneficial effects are neither supported nor excluded by available evidence, but such treatments may cause adverse effects
Rasmussen’s encephalitis				
Gadian J et al. [50]	2017	Systematic review without meta-analysis	To evaluate the effectiveness of IVIG, tacrolimus, corticosteroids, PLEX, ACTH, interferon, and surgery, alone or in combination, in terms of recovery in patients with Rasmussen’s encephalitis	It is possible that IVIG reduces or stops seizures (level 4) IVIG and tacrolimus may be equally effective (level 2b) IVIG or tacrolimus should be taken into consideration for seizure control in Rasmussen’s syndrome (grade C)
Sydenham’s chorea				
Orsini A et al. [59]	2022	Multicenter cohort study	To provide high-quality evidence of the management and prognostic markers of Sydenham’s chorea in 171 pediatric patients	Immunomodulatory therapy (IVIG and steroids) is not associated with a medium-term higher efficacy, but it shows a slightly reduced risk of relapse in comparison with symptomatic therapy (anti-seizure medication, dopamine antagonists)
Mohammad SS et al. [60,61]	2015	Systematic review	To compare effectiveness of IVIG, corticosteroids, PLEX and symptomatic treatment in terms of chorea score reduction and of shorter need for symptomatic treatment considering two randomized controlled studies	A mean chorea score reduction was registered in 29% of prednisone group members, in 72% of IVIG group members and in 50% of PLEX group members The symptomatic treatment combined with IVIG, in comparison with the symptomatic treatment alone, was associated with a significant improvement in clinical score at 30, 90, and 180 days and with the need for a significantly quicker symptomatic treatment
Hematologic conditions				
Acquired hemophilia (AHA)				
Gianella-Borradori et al. [62]	1984	Case report	To describe a case of AHA in a boy of 13 years	Vincristine and steroids controlled the hemophilia, but this therapy was discontinued due to side effects. For this reason, he had to start therapy with IVIG Treatment with IVIG resulted in a slow rise in factor VIII and in a reduction in factor VIII inhibitors
Sultan Y et al. [63]	1984	Case series	To describe the cases of two patients with high-titer antihemophilic factor (VIII) autoantibodies, treated with IVIG	The treatment with IVIG produced rapid and durable antibody suppression
Autoimmune hemolytic anemia (AIHA)				
Flores G et al. [64]	1993	Separate pilot studies at 3 institutions (37 patients), combined with a review of 36 cases of AIHA treated with IVIG reported in the literature	To assess warm antibody AIHA responsiveness to IVIG	Overall, 29 of 73 (39.7%) patients were responsive to IVIG treatment Hepatomegaly and a low level of hemoglobin pre-treatment (<6–7 g/dL) strongly correlated with IVIG efficacy
Mathai et al. [65]	1990	Case report	To describe IVIG efficacy in the management of hemolytic anemia through the case of a newborn with hemolytic anemia refractory to steroid treatment	An excellent response was detected in the patient, achieving complete remission with IVIG (dose of 2 g/kg over 5 days)
Autoimmune neutropenia (AIN)				
Hilgartner et al. [66]	1987	Case series	To describe the cases of six children aged <2 years presenting with a neutrophil count less than 300/mmc and severe infections, treated with 1 g/kg of IVIG until their neutrophil counts reached more than 1000/mmc	Efficacy was reached with a dose of 3 g/kg given over 5 to 7 days; all children recovered from their infections and reached persistent normal neutrophil counts
Bussel et al. [67]	1983	Case report	To describe two cases of children affected by AIN who showed a good response after IVIG treatment	With the initial IVIG therapy, the neutrophil count raised significantly in both patients; one patient achieved remission
Immune thrombocytopenia (ITP)				
Mikhail D. et al. [68]	2022	Monocentric retrospective study	To calculate the rate of rise in the platelet count with IVIG therapy and the impact of demographic and clinical characteristics on the rate of rise. To provide a nomogram showing when to expect a meaningful rise in the platelet count after IVIG therapy	A rate of rise of over +0.5 K/µL/h was seen in 78% of patients. There was not a statistically significant correlation between the rate of rise in the platelet count and age or sex
Heitink-Pollé KMJ et al. [69]	2018	Multicenter randomized controlled trial	To demonstrate IVIG efficacy at a single dose in reducing the chronicization rate in children with the first episode of ITP. 206 children with new-onset ITP, <20 × 10^9^ platelets/L and mild to moderate bleeding randomly received either a single IVIG infusion (0.8 g/kg) (*n* = 102) or no therapy (n = 104)	Chronic ITP interested 18.6% of the IVIG group and 28.9% of the counterpart (relative risk 0.64). IVIG group showed a significantly higher complete response rate in the first 3 months
Higashide Y et al. [70]	2018	Multicentered retrospective study	To identify new predictive factors for determining responses to IVIG treatment in newly diagnosed ITP.	Age ≥23 months (*p* = 0.020) and platelet count <9.0 × 10^9^/L (*p* = 0.018) were considered to be unfavorable factors for short-term response. Age ≥23 months (*p* = 0.020) was the only unfavorable factor for long-term response to IVIG therapy
Ou CY et al. [71]	2006	Comparative study	To compare the increase in platelet count and to assess the risk for persistent severe thrombocytopenia in 87 children with ITP treated with IVIG or prednisolone. 54 patients received IVIG for two days, followed by prednisolone, while 33 patients received prednisolone for 14–21 days	The 81.5% of the children receiving IVIG demonstrated a prompt platelet count increase on the second day, compared to the 39.4% in the prednisolone group (*p* < 0.01) Mean platelet counts did not significantly differ between the 2 groups on the third, fifth and seventh day and on the next days (all *p* > 0.05)
Gereige RS et al. [72]	2000	Retrospective study	To compare the effectiveness of 3 regimens: IVIG only, HD methylprednisolone (HDMP) only and IVIG and HDMP in association in the management of 148 children of ITP	IVIG and IVIG plus HDMP in association were better than HDMP alone in increasing the platelet count by the first 24 h The combination of IVIG and HDMP was statistically superior for the treatment of patients with more than 10.000 × 10^9^ platelets/L than IVIG or HDMP alone IVIG was the least effective in patients with < 10.000 × 10^9^ platelets/L
Fujisawa K et al. [73]	2000	Prospective randomized trial	To determine the minimal essential treatment for acute ITP in patients with <10 × 10 ^3^ platelets/mmc or 10 to 29 × 10 ^3^ platelets/mmc and mucosal bleeding. Patients randomly received IVIG at 1 to 2 g/kg, conventional oral prednisolone, intravenous methylprednisolone (MPSL), or pulsed intravenous methylprednisolone (P-MPSL)	IVIG offered faster platelet enhancement compared with other therapies
Rosthøj S et al. [74]	1998	Randomized study	To compare IVIG with P-MPSL in 43 new-onset pediatric cases of ITP	A partial remission with >50 × 10^9^ platelets/L after 3 days was significantly higher in the IVIG group (21/23) in respect to the counterpart (10/20) receiving MPSL (*p* = 0.003); the mean platelet count after 3 days was 188 in the IVIG group and 77 × 10^9^/L in the counterpart (*p* < 0.001). After six days, the complete response was higher in the IVIG group (16/23) in respect to the counterpart (10/20) (*p* = 0.16) These results show that IVIG infusions are more effective than HD steroids as a first therapy for ITP
Neonatal alloimmune thrombocytopenia (NAIT)				
Winkelhorst D et al. [75]	2017	Review	An overview about the available evidence for both pre- and post-natal treatment alternatives in NAIT	For the postnatal treatment of NAIT, matched platelet transfusion represents a potential treatment, which may be combined with IVIG
Mueller C et al. [76]	1989	Case series	To describe 12 cases of NAIT treated with IVIG.	Of the 12 infants, 10 had a good response (response rate 70–80%), confirming that IVIG can represent a rescue therapy in NAIT in case of unavailable compatible platelets
Post-transfusion purpura (PTP)				
Ziman A et al. [77]	2002	Case report	To describe the case of a female of 61 years of age who developed severe PTP	The patient was effectively treated with IVIG
Kroll H et al. [78]	1993	Retrospective study	To analyze clinical and therapeutic data regarding 38 adult patients with PTP	Of the patients, 74% were responsive to IVIG
Mueller C et al. [79]	1988	Retrospective multicenter study	To assess IVIG efficacy in 19 PTP cases in adults	A good or excellent response was detected in 17/19 patients receiving IVIG
Thrombotic thrombocytopenic purpura (TTP)				
Ding J et al. [80]	2017	Retrospective registry study	To examinate IVIG response in patients with refractory TTP	Of eleven adult patients receiving IVIG due to clinical worsening despite PLEX, six (54.5%)were responsive to IVIG Four of the six patients who initiated IVIG therapy because of new neurological complications reached clinical improvement
Centurioni R. et al. [81]	1995	Retrospective observational study	To demonstrate the role of IVIG in the treatment of TTP through the analysis of 17 cases of PLEX-resistant TTP treated with IVIG	Of 17 patients, 7patients died, while 2/17 and 8/17, respectively, showed a partial and complete remission Only in four of the ten cases with a positive response did the addition of IVIG seem to have had a significant improvement
Inflammatory conditions				
Henoch–Schönlein purpura (HSP)				
Zhang X et al. [82]	2022	Single-center retrospective cohort study	Clinical trial on 64 HSP patients with refractory gastrointestinal manifestation, divided into three groups with different treatment strategies: IVIG and steroid; hemoperfusion and steroid; and IVIG, hemoperfusion and steroid.	Improvement in gastrointestinal symptoms was found in all three groups. IVIG treatment showed better results in young children.
Morotti F et al. [83]	2020	Case reports	Two clinical cases (11- and 5-year-old boys) of severe HSP gastrointestinal vasculitis refractory to HD intravenous corticosteroids	Rapid clinical gastrointestinal response to IVIG therapy, observing complete resolution within a few days
Mauro A et al. [84]	2019	Case report	Clinical case of HSP showing severe cutaneous lesions refractory to corticosteroid therapy	An 11-year-old girl was successfully cured with HD-IVIG with prompt symptom recovery
Cherqaoui B et al. [85]	2016	Retrospective study	Study on the outcomes of HSP patients showing severe gastrointestinal manifestations and undergoing nonsteroidal immunomodulatory treatments	There were eight children (five boys and three girls, 3–15 years old) with severe gastrointestinal involvement. Steroids and IVIG were administered to all of the children included in the study Complete response in 6 within 7 days, partial response in 2 days There were two relapses showing reduction in the gastrointestinal involvement. A second dose of IVIG was administered with a good clinical response
Fatima A et al. [86]	2014	Case report	Clinical case of a 2-year-old girl with HSP showing gastrointestinal involvement (small bowel thickening, colic pneumatosis)	Abdominal pain improved after total parenteral nutrition, antibiotics, and HD-IVIG. Computed tomography scan confirmed normalization of the radiologic signs
De Maddi F et al. [87]	2013	Case report	Clinical report of boy of 8 years of age with HSP (purpuric rash, polyarticular arthralgias, abdominal pain) who developed acute cerebral hemorrhage	After IVIG infusion (combined with methylprednisolone, mannitol, and phenobarbital) they observed improvement in skin lesions, abdominal pain, and a partial resolution of his weakness Two weeks later, neurologic examination confirmed the absence of focal deficit. Improvement in the initial brain lesion at the follow-up MRI
Fagbemi AA et al. [88]	2007	Case report	Clinical case of a 9-year-old affected by HSP, with life-threatening gastrointestinal hemorrhage from multiple intestinal sites. Endoscopy demonstrated aspects of severe gastritis and corticosteroids use was precluded	IVIG use was beneficial for a sustained remission of clinical manifestations. Therefore, this treatment could represent an effective alternative to corticosteroids in these cases
Lamireau T et al. [89]	2001	Case report	Clinical report of a 10-year-old boy with HSP experiencing sever persistent abdominal pain and bleeding 2 months after diagnosis	After HD-IVIG gastrointestinal manifestations resolved within 3 days, feeding via mouth began on day 7, and he stopped steroid therapy within 1 month. No symptoms were reported during a 3-year follow-up
Juvenile dermatomyositis (JDM)				
Aggarwal R et al. [90]	2022	Clinical trial	Randomized, placebo-controlled trial on 95 adult patients showing active dermatomyositis, divided in a 1:1 ratio and receiving IVIG (2 g/kg) or placebo, every 4 weeks for 16 weeks.	Disease improvement (based on a clinical score) was significantly higher in the IVIG-treated patients, despite adverse events
Lam CG et al. [91]	2011	Retrospective study	Study to define if patients with JDM who receive IVIG reach quiescence sooner with a reduced disease activity compared to controls. For SD patients, to compare IVIG-treated patients to IVIG-naive ones and define disease activity in the two groups	Similar or lower disease activity was noted among IVIG patients compared to controls from 30 days to 4 years post-diagnosis. Steroid-resistant patients showed the best response. Among steroid-dependent patients, IVIG recipients had lower disease activity
Levy et al. [92]	2010	Retrospective study	Clinical course of eight patients with JDM cured successfully without the use of systemic corticosteroids	IVIG was the first-line treatment (75%) alongside methotrexate (50%), with a favorable response in all and no serious side effects
Al-Mayouf SM et al. [93]	2000	Retrospective study	To evaluate the role of IVIG in the management of JDM in 18 steroid-resistant or steroid-dependent patients or with unacceptable side effects. Ten patients underwent second-line treatments (methotrexate, azathioprine, cyclosporine, or cyclophosphamide)	Ten patients with clinical improvement and reduction in corticosteroid dose for more than 3 months, without disease flare ups Nine of these patients had IVIG alone as a second-line treatment, whereas three patients were treated with additional therapy. Six patients required multiple agents to reduce JDM activity
Sansome A et al. [94]	1995	Case series	Evaluation of IVIG therapy (2 g/kg for 3 days, then same dose in 5 consecutive days) in nine children with JDM, failing conventional treatments or with intolerable side effects, followed for a 4-year period. The patients were treated with prednisolone, azathioprine, and cyclosporin	Clinical improvement in all nine IVIG-treated patients Among the eight children on prednisolone, the steroid dose was reduced in six and remained the same in two
Collet et al. [95]	1994	Case report	Two patients (7-year-old and 12-year-old) with refractory JDM and steroid toxicity, treated with HD-IVIG for 9 months	Improvement in muscle mass and reduced muscular damage, cutaneous lesions, discontinuation of steroid therapy
Dalakas MC et al. [96]	1993	Randomized controlled trial	Evaluation of IVIG treatment in 15 patients (18 to 55 years of age) diagnosed with resistant dermatomyositis diagnosed through biopsy. Randomization between HD-IVIG or placebo per month for three months while taking steroids, with the possibility of crossing over to the alternative therapy for three months in addition	Improvement in muscle strength and repeated muscular biopsies, neuromuscular symptoms
Lang BA et al. [97]	1991	Pilot study	Evaluation of HD-IVIG in 5 JDM patients, with persistent muscular weakness despite daily corticosteroids. Toxicity to steroid or immunosuppressive therapy was reported by some of the patients	Improved muscle mass and reduced cutaneous rash in all patients. Discontinuation or reduction in steroid therapy
Roifman et al. [98]	1987	Case report	Evaluation of response to IVIG treatment in a patient of 15 years of age presenting with diffuse myositis and normal immunity, not responding to treatment with steroids, methotrexate, or cyclophosphamide	Improvement in muscle strength, cardiac ejection fraction, pulmonary function tests; reduction in steroid dosage
Kawasaki disease (KD)				
Broderick C et al. [99]	2023	Systematic review with meta analysis	To assess IVIG safety and efficacy in the treatment and prevention of KD cardiac complications	The evidence indicates that HD-IVIG may reduce the risk of coronary artery aneurysm formation in comparison with acetylsalicylic acid or medium- or low-dose IVIG Adverse effects incidence is not clinically significantly different in comparison with other treatments Compared to acetylsalicylic acid, HD-IVIG may reduce fever duration, but the need for adjunctive therapy was little or not different between the 2 treatments HD-IVIG probably reduces fever duration and need for adjunctive therapy in comparison with medium- or low-dose IVIG
Lei WT et al. [100]	2022	Network meta-analysis	To make a comparison about the efficacy and safety of the alternative treatments for the initial and refractory stages of KD, examining 56 randomized controlled trials involving 6486 participants	Medium-dosage IVIG (1 g) + aspirin + infliximab was compared with high-dosage IVIG (2 g min) + aspirin, showing a shorter fever duration (mean difference = −1.8 days) and a smaller incidence of coronary artery anomalies in the initial-stage KD (odds ratio = 0.50, 95% confidence interval: 0.18–1.37) High-dosage IVIG (2 g min) + pulse steroid in comparison with high-dosage IVIG (2 g min) alone showed a better rate of fever decline in refractory-stage KD (odds ratio = 0.04, 95% confidence interval: 0.00–0.43). High-dosage IVIG (2 g min) + ciclosporin in respect to high-dosage IVIG alone (2 g min) exhibited a smaller incidence of coronary artery anomalies in refractory-stage KD (odds ratio = 0.05, 95% confidence interval: 0.00–1.21) Infliximab in comparison with high-dosage IVIG (2 g min) alone improved resolution in refractory-stage KD (odds ratio = 0.20, 95% confidence interval: 0.07–0.62)
Huang H et al. [101]	2022	Retrospective and prospective cohort study	To develop a nomogram predictive for IVIG resistance enrolling retrospectively 1293 children with KD in eastern China, with IVIG resistance defined as persistent or recrudescent fever ≥36 h after the stop of IVIG infusion	Low haemoglobin, high percentage of neutrophils, high C-reactive protein level, high platelet count, low serum albumin, high serum sodium, high serum alkaline phosphatase, low coronary artery damage, and uncomplete Kawasaki disease represent risk factors for IVIG resistance. The nomogram based on these factors showed a good discriminatory power (area under curve (AUC), 0.75), sensitivity (0.74) and specificity (0.64). In the prospective (209 patients) and external (205 patients) data, the AUC was 0.83 and 0.66, respectively, the sensitivity was 1 and 0.86, respectively, and the specificity was 0.60 and 0.49, respectively.
Hamada H et al. [102]	2019	Randomized controlled, open-label, blinded-endpoints, phase 3 trial	To compare safety and efficacy of IVIG + ciclosporin (5 mg/kg per day for 5 days) and IVIG in 175 KD patients at high risk for IVIG resistance according to Kobayashi’s scoring system (PMID: 16735679). The primary outcome was incidence of coronary artery anomalies during the 3-month trial.	Incidence of coronary artery anomalies was lower in the IVIG + ciclosporin group than in the IVIG group—with a significant risk ratio (0.46; 95% confidence interval 0.25–0.86; *p* = 0.010)—in KD patients at high risk for IVIG resistance, with no difference as concerns incidence of adverse effects
Dionne A et al. [103]	2018	Monocentric retrospective cohort study	To compare 154 KD children with and without a concomitant infection in terms of need for a second IVIG course, C-reactive protein level, and prevalence of coronary complications	KD children with a concomitant infection showed a higher prevalence of IVIG resistance (*p* = 0.05), of need for a second IVIG course (*p* = 0.04), and higher inflammatory markers prior to (*p* = 0.04) and after (*p* = 0.003) IVIG administration. Prevalence of coronary complications was not statistically different between the two groups
Kaya Akca U et al. [104]	2018	Monocentric retrospective cohort study	To compare different prognostic scoring system predictive for IVIG resistance in Turkish children with KD	Low platelet count and increased serum gamma-glutamyl transferase levels were unfavorable factors for response to IVIG therapy
Lin MT et al. [105]	2015	Retrospective cohort study	Evaluation of IVIG effectiveness in terms of coronary heart complication reversal in patients diagnosed with KD between the groups with and without IVIG	Insufficient evidence about the effectiveness of IVIG in KD in terms of coronary heart complication reversal, with a significant improvement in the early subgroup but not in the late subgroup
Newburger JW et al. [106]	1986	Multicenter randomized trial	Comparison between IVIG + aspirin and aspirin only in terms of prevalence of coronary–artery abnormalities at a timepoint of 2 and 7 weeks after enrollment	IVIG safely and effectively reduces, in a statistically significant manner, the rate of coronary–artery abnormalities if administered early in the course of KD
Furusho K et al. [107]	1983	Case–control study	Comparison between IVIG and aspirin in terms of duration of fever, time to phlogosis indexes negativization, and aneurysm formation	IVIG prevents coronary aneurysm formation, providing an anti-inflammatory effect reducing significantly fever duration and time to negativization of phlogosis indexes
Multisystem inflammatory syndrome in children (MIS-C)				
Mahmoud S et al. [108]	2022	Review	An overview about the available evidence and recommendations for MIS-C treatment in children and adolescents	IVIG ± glucocorticoids represent the first-tier immunomodulatory therapy. In case of refractoriness, a switch to biologic agents such as anakinra, tocilizumab, and infliximab is warranted
Son MB et al. [109]	2021	Comparative study	To compare the effectiveness of IVIG + steroids and IVIG alone in 518 patients with MIS-C <21 years of life, considering cardiac dysfunction in terms of need for vasopressors on or after day 2 of therapy as primary outcome and adjunctive treatment (glucocorticoids in patients receiving IVIG alone, a second course of IVIG, or a biologic) on or after day 1, and fever on or after day 2, peristent or recurrent, as secondary outcomes	IVIG + steroids therapy carries a lower risk of new or persistent cardiac dysfunction than IVIG only.
Ouldali N et al. [110]	2021	Retrospective cohort study	Comparison between IVIG + methylprednisolone and IVIG alone in terms of fever duration	IVIG + methylprednisolone treatment significantly reduces rate of unresponsiveness, described as persistent fever after 48 h of first-line therapy or fever relapse within a week
Belhadjer Z et al. [111]	2020	Observational study	Comparison between IVIG + methylprednisolone and IVIG alone in terms of recovery of cardiac function	IVIG + methylprednisolone treatment significantly reduces time to recovery, described as isovolumic relaxation time >90 milliseconds for diastolic function and left ventricle ejection fraction >55%
Polyarthritis (rheumatoid factor positive)				
Giannini EH et al. [112]	1996	Clinical trial (randomized, withdrawal, double-blind, placebo-controlled)	Evaluation about efficacy and safety of IVIG therapy in the treatment of resistant polyarticular juvenile idiopathic arthritis	Three-fourths of patients with poly-JIA experienced clinical improvement with IVIG during open administration, more likely in those with a history of disease shorter than 3 years than those with a more prolonged illness (>5 years). The beneficial effect was short after discontinuation of IVIG therapy
Systemic juvenile idiopathic arthritis (JIA)				
Uziel Y et al. [113]	1996	Clinical trial (open-label)	To assess the short- and long-term consequences of IVIG therapy among 27 patients affected by systemic onset juvenile idiopathic arthritis (SOJIA)	Clinical improvement in systemic symptoms (decrease in fever, significant reduction in the steroid dose). Less predictable effects on the control of arthritis IVIG showed a limited effect on patients with severe forms of SOJIA and with a history of resistance to standard therapy
Silverman ED et al. [114]	1994	Clinical trial (multicenter, randomized, double-blind, placebo-controlled)	Evaluation of safety and efficacy of IVIG therapy in 31 patients with juvenile idiopathic arthritis (dosage of 1.5 g/kg every 2 weeks for 2 months, then every month for 4 months). Comparison with placebo (0.1% albumin)	No effect of IVIG in reducing fever and other systemic manifestations, or on joint count, hemoglobin, albumin, platelet count, and erythrocyte sedimentation rate was noticed, compared to placebo Results are nondefinitive because of the low number of enrolled patients
Silverman ED et al. [115]	1990	Clinical trial (open-label)	Evaluation of HD-IVIG treatment in eight patients affected by systemic juvenile idiopathic arthritis considering articular and extraarticular symptoms, laboratory exams, corticosteroid therapy, and global evaluation	Improvement in articular and extraarticular symptoms; prednisone dosage; blood exams (hemoglobin, albumin, serum immunoglobulin, platelet count, and erythrocyte sedimentation rate levels)
Prieur AM et al. [116]	1990	Clinical trial (open-label)	Evaluation of HD-IVIG treatment in 16 patients with severe forms of juvenile chronic arthritis	Improvement in laboratory abnormalities. No difference among short- and long-term therapy
Systemic lupus erythematosus (SLE)				
Shi N et al. [117]	2021	Case report	Clinical case of an 11-year-old girl presenting Childhood-Onset SLE (cSLE)-associated MAS and neuroimaging anomalies as onset symptoms, treated with HD methylprednisolone, and cyclophosphamide and HD-IVIG every two weeks.	The patient showed clinical and neuroimaging improvement on day 25.
Akca U et al. [118]	2021	Retrospective multicenter study	To assess hematological features in pediatric SLE patients	215 children with SLE were studied. Of 118 children with hematological involvement, the principal treatment consisted of corticosteroids followed by IVIG Second-line therapy for resistant patients was rituximab
Baglan E et al. [119]	2021	Case report	Clinical case of a boy of 10 years of age with cSLE showing chorea and delirium, who underwent Zipper method treatment (consecutive PLEX and IVIG) given the resistance to conventional therapy and rapidly progressing neurological involvement	First cSLE case involving central nervous system, showing resistance to steroids and cyclophosphamide, and resolved by the Zipper method
Lube GE et al. [120]	2016	Retrospective multicenter cohort study	Evaluation of prevalence, clinical manifestations, laboratory exams, and outcomes in a childhood-onset SLE cohort with Evans Syndrome at diagnosis	Evans Syndrome was observed in 11 of 850 (1.3%) childhood-onset SLE patients The prevalence of intravenous steroids and IVIG between childhood-onset SLE patients with Evans syndrome at diagnosis was statistically superior in the first cohort
Brogna C et al. [121]	2016	Case report	Clinical report of a boy of 7 years of age showing severe neuropsychiatric lupus erythematosus provoked by reactivation of EBV cerebral infection, effectively responding to IVIG	The patient presented severe neuropsychiatric features refractory to standard therapy with psychotropic medications and corticosteroids They observed a good response of central symptoms to IVIG treatment, with recovery of neurological signs and reduction in SLE antibody
Friedman DL et al. [122]	2010	Prospective multicenter open-label study	To assess the function of IVIG as prevention treatment for congenital heart block, given its potential role in reducing maternal autoantibody levels and fetal inflammatory activation	IVIG protocol was completed in twenty women before the limit of 3 cases of advanced congenital heart block was observed. No signs of neonatal lupus were observed in sixteen children at birth. No substantial variations in maternal antibody levels were noticed. There were no safety issues
Pisoni CL et al. [123]	2010	Prospective multicenter observational study	Evaluation of IVIG therapy in congenital heart block prevention among neonates born from high-risk pregnancies	IVIG therapy (dosed as follows: 400 mg/kg at weeks 12, 15, 18, 21, and 24 of pregnancy) did not demonstrate any effect as a prophylaxis for congenital heart block in high-risk women
Miyagawa S et al. [124]	2000	Case report	Clinical case of a girl of 7 years of age with SLE and probable diagnosis of Sjögren’s syndrome who developed Guillain-Barré syndrome 6 years after diagnosis	The patient condition resolved with aggressive treatment, involving HD steroids, PLEX, and IVIG therapy

**Table 2 cells-12-02417-t002:** IVIG-related major adverse events and the relative monitoring, preventive, and therapeutic strategies in a pediatric context.

Major Adverse Event	Monitoring and Treatment	Preventive Measures	Cost-Effective Commercial Preparations *
Acute renal failure [237,238,239]	Monitoring: Urine output, creatinine, and blood urea nitrogen dosage before and after IVIG employment. Treatment: Immediate IVIG cessation. Intravenous hydration post infusion. Dialysis	Avoidance of sucrose-stabilized (e.g., Carimune, Cytogam, Gammar-IV, Sandoglobulin IVIG) and/or hyperosmolar (e.g., Asceniv, Bivigam, Gammagard S/D, Gammaplex 5%, Venoglobulin-I 10% IVIG; Cutaquig or Xembify SCIG) products, especially in case of concomitant nephrotoxic therapies (e.g., inhibitors of the renin-angiotensin system, NSAIDs), diabetes mellitus, pre-existing renal disease, sepsis, hypovolemia. Prehydration	CUVITRU SCIG 200 mg/mL 5 g: $459.54 = 415.90 EUR Hizentra SCIG 200 mg/mL 5 g: $500.99 = 453.86 EUR HYQVIA SCIG 100 mg/mL 5 g: $500.99 = 453.86 EUR Privigen IVIG 100 mg/mL 5 g: $500.99 = 453.86 EUR Gammagard IVIG 5 g/96 mL 5 g: $500.99 = 453.86 EUR Flebogamma IVIG 50 mg/mL 5 g: $500.99 = 453.86 EUR Octagam 5% IVIG 50 mg/mL 5 g: $500.99 = 453.86 EUR Gamunex-C IVIG/SCIG 5 g/50 mL 5 g: $768.35 = 695.37 EUR Gammagard liquid SCIG 5 g/50 mL 5 g: $897.55 = 812.30 EUR Gammaked IVIG/SCIG 1 g /10 mL 5 g: $998.32 = 904.01 EUR PANZYGA IVIG 100 mg/mL 5 g: $1087.88 = 985.20 EUR Gammaplex 10% IVIG 50 mg/mL 5 g: $1091.09 = 988.38 EUR Gamimune N 5%–10% IVIG (out of production) Venoglobulin-I 5% IVIG (out of production) Venoglobulin-S IVIG (out of production)
Anaphylaxis [238,239,240,241]	Treatment: Immediate IVIG cessation.	Employment of IgA-depleted products (e.g., Gammagard S/D or Gammaplex IVIG; Cutaquig or Xembify SCIG even better), although the presence of anti-IgA antibodies is neither a necessary nor a sufficient cause of transfusion-related anaphylaxis	Cutaquig SCIG 165 mg/mL 5 g: $637.62 = 577.64 EUR Xembify SCIG 1 g/5 mL–4 g/20 mL 5 g: $946.69 = 857.17 EUR Gammaplex IVIG 50 mg/mL 5 g: $1091.09 = 988.38 EUR Gammagard S/D IVIG 5 g/96 mL 5 g: $1144.79 = 1036.08 EUR
Hemolytic anemia [125,238,239]	Monitoring: Hemoglobin dosage up to 96 h after IVIG. Treatment: Immediate IVIG cessation. Adequate hydration. Red blood cell transfusions. B12 and folate supplementation. Plasmapheresis or hemodialysis as rescue therapies.	Avoidance of amino acid-stabilized products (e.g., Asceniv, Bivigam, Gammagard Liquid-S/D, Gammaked, Gammaplex 10%, Gamunex-C, PANZYGA or Privigen IVIG; CUVITRU, Gammagard Liquid, Gammaked, Gamunex-C, Hizentra, HYQVIA, Xembify SCIG), especially in the case of non-O blood group recipients or concomitant assumption of NSAIDs, gastro-intestinal medications, and broad-spectrum antibiotics. Prehydration	Flebogamma IVIG 50 mg/mL 5 g: $500.99 = 453.86 EUR Octagam 5% IVIG 50 mg/mL 5 g: $500.99 = 453.86 EUR Cutaquig SCIG 165 mg/mL 5 g: $637.62 = 577.64 EUR Gammaplex 5% IVIG 50 mg/mL 5 g: $1091.09 = 988.38 EUR Cytogam IVIG 50 mg/mL 5 g: $3718.92 = 3368.65 EUR Carimune IVIG (out of production) Gamimune N 5% IVIG (out of production) Gammar-IV IVIG (out of production) Venoglobulin-I IVIG (out of production) Venoglobulin-S IVIG (out of production)
Myocardial infarction, coronary spastic angina (reported only in non-pediatric patients) [238,239,242,243]	Treatment: Immediate IVIG cessation.	Avoidance of high-sodium-containing (e.g., Asceniv, Bivigam, Gammagard S/D, Gammaplex 5%, Venoglobulin-I 10%, Gammar-IV 10% IVIG) and hyperosmolar (e.g., Carimune 6%–12% IVIG; Cutaquig or Xembify SCIG) products, particularly in case of pre-existent cardiac dysfunction and/or hypertension.	Flebogamma IVIG 50 mg/mL 5 g: $500.99 = 453.86 EUR Octagam 5% IVIG 50 mg/mL 5 g: $500.99 = 453.86 EUR Gamunex-C IVIG 100 mg/mL 5 g: $730.38 = 661.02 EUR Gammagard liquid IVIG 5 g/50 mL 5 g: $897.55 = 812.30 EUR Gammaked IVIG 1 g/10 mL 5 g: $998.32 = 904.01 EUR PANZYGA IVIG 100 mg/mL 5 g: $1087.88 = 985.20 EUR Gammaplex 10% IVIG 50 mg/mL 5 g: $1091.09 = 988.38 EUR Cytogam IVIG 50 mg/mL 5 g: $3718.92 = 3368.65 EUR Carimune 3% IVIG (out of production) Gamimune N 5%–10% IVIG (out of production) Gammar-IV 5% IVIG (out of production) Venoglobulin-I 5% IVIG (out of production) Venoglobulin-S IVIG (out of production)
Thromboembolic events [238,239,244]	Monitoring: Pre and post infusion D-dimer levels Treatment: Immediate IVIG cessation or dose fragmentation and/or lower infusion rate. Discuss low-molecular-weight heparin (1 mg/kg/dose twice a day at an anticoagulation dose). Interruption of pro-thrombotic agents (e.g., oral contraceptive).	Removal of pro-thrombotic factors, such as obesity and oral contraceptives.	
Transfusion-related acute lung injury (TRALI) [238,239,245,246,247]	Treatment: Immediate IVIG cessation. Oxygen supply. Ventilation. Extracorporeal membrane oxygenation. Intravenous ascorbic acid (2.5 g/6 h for 96 h).	No robust evidence available on preventive strategies, although leukoreduction and pathogen reduction technologies, removing membrane lipid structures, and pathogenic nucleic acids in donor samples could have a role in preventing TRALI	

* Commercial preparations are reported in a decrescent order of cost, with price being expressed per 5 g of product; in case of equal price, we reported first the more concentrated preparation.

## Data Availability

The data supporting the findings of this study are available in the figure and tables of this article.

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
