# Peer review of "Anti-Inflammatory and Immunomodulatory Effect of High-Dose Immunoglobulins in Children: From Approved Indications to Off-Label Use"

_cells, 2023, doi:10.3390/cells12192417_

Round 1
Reviewer 1 Report
This article is the first comprehensive review of anti-inflammatory and immunomodulatory effects of high-dose immunoglobulins (HDIG) in children. It provides all aspects of HDIG application in various pediatric immune dysregulation disorders, including hematologic, neurologic, rheumatologic and hyperinflammatory; the mechanisms, benefits, cost-effectiveness, and administration routes of HDIG; monitoring, prevention, and treatment of adverse effects; comparisons to other immune therapies. Thus, it provides information needed for clinicians to use HDIG effectively and responsibly.
Monir comment:
Figure 1: Please make autoantibody and Fab different colors to be distinguishable and label all cell types in the figure to make it easy to follow.
Author Response
Dear Revisor, thank you very much for this valuable observation. We tried to improve the figure making the suggested changes.
Reviewer 2 Report
The manuscript discusses the historical use of immunoglobulins in patients with inborn errors of immunity (IEIs) and the recent imbalance between their increasing demand and availability, exacerbated by the SARS-CoV-2 pandemic. The review aims to provide practical guidance for the need-based use of high-dose immunoglobulins in pediatric patients with immune-dysregulation disorders, emphasizing the importance of responsible utilization, cost-effectiveness, and further research in this area. Nonetheless, there are areas within the manuscript that could benefit from improvement and clarification to enhance its quality.
- The introduction sets the stage for the manuscript but is quite lengthy. Consider condensing it to provide a more concise background on the use of HD-IVIG in pediatric patients. Focus on the key milestones and significance of this therapy.
- The Materials and Methods briefly mentions a literature search but lacks details on the search strategy, inclusion/exclusion criteria, and the number of articles reviewed. Consider providing a more comprehensive description of the research methodology.
- The manuscript frequently references previous studies and guidelines, but it would benefit from in-text citations for specific claims or statements. This would improve the clarity and credibility of the information presented.
- The manuscript briefly mentions the need for responsible use of immunoglobulins due to their limited availability. Consider expanding on the implications of this scarcity and providing recommendations or suggestions for healthcare professionals regarding the appropriate and ethical use of HD-IVIG.
- The manuscript is quite lengthy, and some sections could be condensed for brevity. Focus on conveying essential information while maintaining a clear and concise writing style.
- Ensure consistent use of terminology throughout the manuscript. For example, consider using either "HD-IVIG" or "high-dose intravenous immunoglobulin" consistently, as appropriate.
NA
Author Response
- The introduction sets the stage for the manuscript but is quite lengthy. Consider condensing it to provide a more concise background on the use of HD-IVIG in pediatric patients. Focus on the key milestones and significance of this therapy.
Thank you very much for this precious comment. As You suggested, we cut some paragraphs describing the reasons for IVIG and SCIG shortage, given that it is largely explained in the discussion, focusing more on HD-IVIG mechanisms of action and label and off-label indications.
- The Materials and Methods briefly mentions a literature search but lacks details on the search strategy, inclusion/exclusion criteria, and the number of articles reviewed. Consider providing a more comprehensive description of the research methodology.
Thank you very much for this valuable observation. As You suggested, we enriched the Materials and Methods section adding 4 paragraphs: Search Strategy, Eligibility Criteria, Study Selection and Results.
- The manuscript frequently references previous studies and guidelines, but it would benefit from in-text citations for specific claims or statements. This would improve the clarity and credibility of the information presented.
Thank you very much for this precious comment. As You suggested, we added in each paragraph in-text citations in brackets for the most relevant studies and guidelines cited in our review.
- The manuscript briefly mentions the need for responsible use of immunoglobulins due to their limited availability. Consider expanding on the implications of this scarcity and providing recommendations or suggestions for healthcare professionals regarding the appropriate and ethical use of HD-IVIG.
Thank you very much for this valuable observation. As You suggested, we expanded the discussion about the consequences of IVIG/SCIG shortage, providing pragmatic key messages to healthcare professionals for a more responsible and ethical employment of this medical product.
- The manuscript is quite lengthy, and some sections could be condensed for brevity. Focus on conveying essential information while maintaining a clear and concise writing style.
Thank you very much for this precious comment. As You suggested, we tried to be more concise, focusing on the key milestones and significance of HD-IVIG/-SCIG therapy for each immune-dysregulatory disorder analysed. In adjunction, we condensed the “rheumatologic disorders” and “hyper-inflammatory conditions” sections, both in the text and in the table, in a single section called “inflammatory diseases”.
- Ensure consistent use of terminology throughout the manuscript. For example, consider using either "HD-IVIG" or "high-dose intravenous immunoglobulin" consistently, as appropriate.
Thank you very much for this valuable observation. As You suggested, we tried to use properly the abbreviations throughout the text.
- Minor editing of English language required.
Thank you very much for this precious comment. As You suggested, we made our manuscript edited by a Native English Speaker, trying to use a correct and more readable English.
